# Efficient Testing for Correlation Clustering: Improved Algorithms and Optimal Bounds

**Chengyuan Deng**[*]    **Jie Gao**[†]    **Songhua He**[‡]    **Chen Wang**[§]

## Abstract

Correlation clustering is an important unsupervised learning problem with broad applications. In this problem, we are given a labeled complete graph $G = (V, E^+ \cup E^-)$, and the optimal clustering is defined as a partition of the vertices that minimizes the $+$ edges between clusters and $-$ edges within clusters. We investigate efficient algorithms to test the *cost* of correlation clustering: here, we want to know whether the graph could be (nearly) perfectly clustered (with $0$ or low cost) or is far away from admitting any perfect clustering. The problem has attracted significant attention aimed at modern large-scale applications, and the state-of-the-art results use $\widetilde{O}(1/\varepsilon^7)$ queries and time (up to log factors)[1] to decide whether a graph is perfectly clusterable or needs to flip labels of $\varepsilon\binom{n}{2}$ edges to become clusterable. In this paper, we improve this bound significantly by designing an algorithm that uses $O(1/\varepsilon^2)$ queries and time. Furthermore, we derive the first algorithm that tests the cost for the special setting of correlation clustering with $k$ clusters with $O(1/\varepsilon^4)$ queries and time for constant $k$. Finally, for the special case of $k = 2$, which corresponds to the strong structure balance problem in social networks, we obtain tight bounds of $\Theta(1/\varepsilon)$ queries – the first set of *tight* bounds in these problems. We conduct experiments on simulated and real-world datasets, and empirical results demonstrate the advantages of our algorithms.

## 1 Introduction

Correlation clustering is a fundamental unsupervised problem that has been studied extensively in the literature of theoretical computer science and machine learning. At a high level, the problem asks to partition the datasets based on *qualitative* information, i.e., whether two data points are similar. More formally, the dataset is represented as a labeled complete graph $G = (V, E^+ \cup E^-)$, where each vertex $v \in V$ represents a data point, and each vertex pair $(u, v)$ contains an edge with label $(+)$ or $(-)$ denoting "similarity" and "dissimilarity". The cost of a clustering is defined as the total number of $(+)$ edges crossing clusters and the number of $(-)$ edges inside the same clusters.

Correlation clustering has a broad range of applications, including document summarization Bansal et al. (2002), image segmentation Kim et al. (2011); Yarkony et al. (2012), bioinformatics Hou et al. (2016), and community detection Veldt et al. (2018); Shi et al. (2021). Notably, correlation clustering corresponds to naturally emerging structures in signed social networks, where edges are classified as "friendly" and "hostile" relationships. In this setting, structural balance theory, which is well established in sociology, characterizes the "stability" of triangles in signed networks Heider (1946; 1982); Cartwright & Harary (1956); Davis (1967). With strong structural balance, only two types of triangles are stable — with all three edges as positive or with two negative and one positive edge ("the enemy of your enemy is your friend"). The local stability condition also implies global alignment. The vertices in a stable signed network can be partitioned into two groups with all $(+)$ (friendly) edges inside each group and all $(-)$ (hostile) edges in between. Mathematically, this is

---

[*]Rutgers University. E-mail: cd751@rutgers.edu.

[†]Rutgers University. E-mail: jg1555@cs.rutgers.edu.

[‡]Rutgers University. E-mail: songhuahe.cs@gmail.com.

[§]Rensselaer Polytechnic Institute. E-mail: wangc33@rpi.edu.

[1]We recently found out that Goldreich and Ron studied the equivalent clique-collection property in the dense graph model. Their results imply tight $\widetilde{\Theta}(1/\varepsilon)$ query bounds for the clusterability and fixed-k clusterability testing problems considered here; see the Addendum on the next page for more details.

precisely the case of a *perfect (zero-cost) clustering* with two clusters. In addition, a weaker version of structural balance also allows triangles of all three negative edges. Globally, a weakly balanced signed network corresponds to multiple clusters with only positive intra-cluster edges and negative inter-cluster edges. That is, the network has a zero cost correlation clustering where the number of clusters can be flexible. In the remainder of this paper, such a network with a zero cost correlation clustering is called *clusterable* (with any number of clusters), *k-clusterable* if the number of clusters is fixed to be $k$. A (strongly) balanced network is 2-clusterable.

Most of the work in correlation clustering (resp. structural balance) aims to find or approximate the best *clustering*, i.e., output a partition of the vertices (see, e.g. Bansal et al. (2002); Ailon et al. (2008); Chawla et al. (2015); Cohen-Addad et al. (2021); Assadi & Wang (2022); Cohen-Addad et al. (2022; 2023); Dalirrooyfard et al. (2024); Cohen-Addad et al. (2024a;b); Cao et al. (2024); Dalirrooyfard et al. (2025), and references therein). For an $n$-vertex graph, simply outputting the partition requires $\Omega(n)$ time. There are efficient algorithms that converge in near-optimal $\widetilde{O}(n)$ time[2] Assadi & Wang (2022); Cao et al. (2024; 2025). Nevertheless, in applications with massive datasets, we might want to learn the *cost* of correlation clustering using $o(n)$ time. For instance, in the structural balance problem, we might be interested in knowing whether the graph is close to or far away from being balanced without knowing the entire network structure. A graph that is far from a balanced state may indicate high level of volatility. Additionally, we might want to use the clustering cost to determine whether the graph is *worthy of clustering* without paying $\Omega(n)$ time.

The above question is closely related to the realm of *property testing*, in which we are often interested in obtaining statistics of the data with only a very limited number of queries. For correlation clustering, a handful of existing results have explored this direction. For instance, Bonchi, García-Soriano, and Kutzkov Bonchi et al. (2013) designed an algorithm that computes a data structure that supports cluster membership query in $O(1/\varepsilon^2)$ time, and the underlying solution is a $3\mathsf{OPT} + \varepsilon n^2$ approximation[3]. Subsequently, Assadi et al. (2023) and Ashvinkumar et al. (2023) studied the problem of testing for the cost of correlation clustering and structural balance in the *streaming* model, where the edges arrive one-by-one in a stream. There, the goal was to obtain an approximation of the optimal clustering cost with $o(n)$ *space*.

To the best of our knowledge, the work closest to the problem for sublinear *time* is Adriaens & Apers (2023) (see also Chen et al. (2024) for the quantum setting), where they designed an algorithm requiring $\widetilde{O}(1/\varepsilon^7)$ queries, to test whether a graph is $\varepsilon/10$-*close-to-clusterable* vs. $\varepsilon$-*far from being clusterable*. With a stronger technique by Sohler (2012), one can use $\widetilde{O}(1/\varepsilon^2)$ queries to test whether a graph is *balanced* vs. $\varepsilon$-*far from being balanced*, for the special case of structural balance. Throughout, $\varepsilon$-far indicates at least $\varepsilon\binom{n}{2}$ edge labels need to be flipped to make the graph balanced or admit a perfect clustering. To date, there are no matching lower bounds to show the tightness of these results, and we do *not* have knowledge on testing correlation clustering cost with $k$ (which is given) clusters for general $k$. Therefore, getting improved bounds, and ideally tight bounds, for testing correlation clustering and structural balance remains important open problems.

**Addendum.** As of April 2026, the authors notice that Goldreich & Ron (2011) studied the clique collection problem in the same model, which is essentially equivalent to the problems studied in this paper. They show a tight bound of $\tilde{\Theta}(1/\varepsilon)$ for the clique collection problem and for its fixed-$k$ version. In addition, they show a tight bound of $\Theta(1/\varepsilon^{4/3})$ for non-adaptive testers. However, our technique and analysis is novel, and we hope they can find applications in other property testing problems.

## 1.1 OUR CONTRIBUTIONS

We make substantial progress towards the open problems in this paper. We consider the model where one can issue queries for the label of any edge $(u, v)$ and we minimize the number of queries used to evaluate or approximate the cost of correlation clustering for the graph. In particular, Our contributions are summarized in the following settings.

---

[2]Unless specified otherwise, we use $\widetilde{O}(\cdot)$ to hide poly-logarithmic terms.

[3]Unless specified otherwise, the notation $\mathsf{OPT}$ denotes the optimal clustering cost for correlation clustering.

- We propose an algorithm to test whether the correlation clustering cost is at most $O(\varepsilon^2 \binom{n}{2})$ or at least $\varepsilon \binom{n}{2}$ using $O(1/\varepsilon^2)$ queries.

- We give an algorithm to test whether the correlation clustering cost with $k$ clusters for any constant $k$ is $\{O(\frac{\varepsilon^4}{k^4 \ln^4 k} \binom{n}{2})$ or at least $\varepsilon \binom{n}{2}$ using $O(1/\varepsilon^4)$ queries.

- For the case of $k = 2$, which corresponds to structural balance, we devise an algorithm that tests if the graph is at most $\varepsilon/900$-close to being balanced or at least $\varepsilon$-far from being balanced using $O(1/\varepsilon)$ queries. We complement the upper bound by an $\Omega(1/\varepsilon)$ lower bound, showing the tightness of the proposed algorithm.

Note that all algorithms are efficient in time complexity as well: it is proportional to the query complexity. Table 1 shows comparison of results. Prior results are from Adriaens & Apers (2023).

Table 1: Comparison of Results on Query Complexity.

| Task | Previous Best Bound | Our U.B. | Our L.B. | Remark |
|---|---|---|---|---|
| Structural Balance | $\widetilde{O}(1/\varepsilon^2)$ | $O(1/\varepsilon)$ | $\Omega(1/\varepsilon)$ | — |
| Correlation Clustering | $\widetilde{O}(1/\varepsilon^7)$ | $O(1/\varepsilon^2)$ | $\Omega(1/\varepsilon)$ | — |
| Correlation Clustering (fixed $k$) | — | $O(1/\varepsilon^4)$ | $\Omega(1/\varepsilon)$ | $O(\frac{k^4 \ln^4 k}{\varepsilon^4})$ for general $k$ |

We now discuss the formal statements for these algorithmic results. We start with the results for testing clusterability for general correlation clustering, which is our main technical result.

**Theorem 1.** *Fix $\varepsilon \in (0, 1)$. There exists a randomized algorithm that given a labeled complete graph $G = (V, E^+ \cup E^-)$ and a parameter $\varepsilon$ answers with the following rules*

- *If $G$ is clusterable, the algorithm always answers "YES";*

- *If $G$ is at least $\varepsilon$-far from being clusterable, the algorithm answers "NO" with probability $\geq 0.9$;*

- *If $G$ is $C \cdot \varepsilon^2$-close to being clusterable for some small constant $C$, the algorithm answers "YES" with probability $\geq 0.9$.*

*The algorithm queries at most $O(1/\varepsilon^2)$ edges of $G$ and runs in $O(1/\varepsilon^2)$ time.*

Compared to the results in Adriaens & Apers (2023), our results improve the query complexity from $\widetilde{O}(1/\varepsilon^7)$ to $O(1/\varepsilon^2)$. Similar to their setting, our algorithm can conduct both vanilla and tolerant property testing. Ignoring the constant factors, for $\varepsilon = 0.01$, the algorithm of Adriaens & Apers (2023) takes $10^{14}$ operations, while our algorithm takes $10^4$ operations. Assuming a machine that takes $10^{-10}$ seconds to process one operation, the running time difference between their algorithm and ours is $> 2.5$ hours vs. less than one second.

Our algorithm is straightforward: we uniformly sample $O(1/\varepsilon)$ vertices and test on their induced subgraph. The analysis rests on a key insight: if a graph is $\varepsilon$-far from being clusterable, this property will be evident even in a small, random sample. We achieve this by introducing Janson's inequality from the random graph theory, which is novel in analyzing property testing algorithms. The proof of Theorem 1 can be found in Appendix E.

We note that our contribution primarily lies in the analysis rather than the design of the algorithm. Results in the literature have shown that property testing problems for graph problems inherently admit relatively simple algorithms Goldreich & Trevisan (2003). Therefore, the crucial and non-trivial part is to conduct better analysis to improve the sample complexity, which is exactly what we did in our paper.

We then investigate the test of clusterability for graphs with $k$ clusters for any integer $k \geq 2$. We obtain the following result.

**Theorem 2.** *Fix $k \geq 2$ and $\varepsilon \in (0, 1)$. There exists a randomized algorithm that given a labeled complete graph $G = (V, E^+ \cup E^-)$ and a parameter $\varepsilon$ answers the following*

- *If $G$ is $k$-clusterable, the algorithm always answers "YES";*

- *If $G$ is at least $\varepsilon$-far from being $k$-clusterable, the algorithm answers "NO" with probability $\geq 0.9$;*

- *In addition, if $G$ is $\left(\frac{\varepsilon^4}{10^{26}k^4 \ln^4 k}\right)$-close-to-$k$-clusterable, the algorithm answers "YES" with probability $\geq 0.9$.*

*The algorithm queries at most $O(\frac{k^4 \ln^4 k}{\varepsilon^4})$ edges of $G$ and runs in $O(\frac{k^4 \ln^4 k}{\varepsilon^4})$ time.*

When $k$ is a constant, the above gives an $O(1/\varepsilon^4)$ algorithm for testing $k$-clusterability.

As far as we are aware, Theorem 2 is the first nontrivial algorithm that tests the clusterability for correlation clustering with $k$ clusters. The dependency of $k$ is polynomial in Theorem 2. By a common observation (see, e.g. Bansal et al. (2002); Adriaens & Apers (2023)), the optimal correlation clustering cost could always be approximated by the optimal solution with at most $O(1/\varepsilon)$ clusters with $O(\varepsilon n^2)$ additive error. Therefore, Theorem 2 also implies testing algorithms with $1/\text{poly}(\varepsilon)$ queries for any meaningful choices of $k$.

Our algorithm for Theorem 2 is a combination of two algorithms: the algorithm used in Theorem 1, and a new algorithm that distinguishes whether a *clusterable* graph is $k$-clusterable or $\varepsilon$-far from $k$-clusterable. We show that the second algorithm works even when the input graph is close enough to be clusterable. Specifically, when the input graph is $\varepsilon^2$-close to a clusterable one, our second algorithm ignores its deviation from its closest clusterable graph with high probability. We prove Theorem 2 formally at Section 3.

We now move on to the special case of $k = 2$, which is mathematically equivalent to the structural balance problem.

**Theorem 3.** *Fix $\varepsilon \in (0, 1)$. There exists a randomized algorithm that given a labeled complete graph $G = (V, E^+ \cup E^-)$ and a parameter $\varepsilon$ answers the following*

- *If $G$ is balanced, the algorithm always answers "YES";*

- *If $G$ is at least $\varepsilon$-far from being balanced, the algorithm answers "NO" with probability $\geq 0.9$.*

*The algorithm queries at most $O(1/\varepsilon)$ edges of $G$ and runs in $O(1/\varepsilon)$ time.*

Theorem 3 uses algorithmic procedures that are fairly different from the subroutines in Theorem 1 and Theorem 2. Here, instead of sampling a subset of vertices and their induced subgraph, we sample *triangles* directly. This in particular avoids the quadratic blow-up in Theorem 1 and Theorem 2.

Similar to our results for general clusterability, our techniques for Theorem 3 extend to *tolerant testing*. However, for structural balance, we obtain stronger guarantees: while the previous algorithms require the graph to be $O(\varepsilon^2)$-close to clusterable, here we can distinguish graphs that are $\delta$-close from being balanced (where $\delta \approx O(\varepsilon)$) versus graphs that are $\varepsilon$-far. We refer to Appendix F and Appendix G for the formal proof.

**Theorem 4.** *Fix $\varepsilon \in (0, 1)$ such that $\delta \leq \varepsilon/900$. There exists a randomized algorithm that given a labeled complete graph $G = (V, E^+ \cup E^-)$ and parameters $\varepsilon, \delta$ answers the following*

- *If $G$ is at most $\delta$-close from being balanced, the algorithm answers "YES" with probability $\geq 0.99$;*

- *If $G$ is at least $\varepsilon$-far from being balanced, the algorithm answers "NO" with probability $\geq 0.99$.*

*The algorithm queries at most $O(1/\varepsilon)$ edges of $G$ and runs in $O(1/\varepsilon)$ time.*

Finally, we present a lower bound result, showing that $\Omega(1/\varepsilon)$ queries are *necessary* to distinguish graphs that are balanced (resp. clusterable) vs. $\varepsilon$-far from being balanced (resp. clusterable).

**Theorem 5.** *Any (possibly randomized) algorithm that given a complete labeled graph $G = (V, E^+ \cup E^-)$, with probability at least $2/3$ answers correctly whether $G$ is balanced or at least $\varepsilon$-far from being balanced requires at least $\Omega(1/\varepsilon)$ edge queries to the graph.*

*Furthmore, the lower bound extends to testing clusterability (for both general $k$ and fixed $k$).*

Theorem 5 indicates that our results for Theorem 3 and Theorem 4 are asymptotically *tight*. To the best of our knowledge, this is the first result that obtained tight bounds in the related literature. The proof can be found at Appendix H.

**Experiments.** We implement the proposed algorithms and evaluate them on synthetic and real-world datasets. Our algorithms demonstrate favorable efficiency in both the query complexity and the running time. For structural balance testing on graphs of size 1000, our algorithm shows a reduction factor of 15 on query complexity and roughly $10^4$ on the running time, comparing to Adriaens & Apers (2023). Our implementation is available on Anonymous Github[4].

**Further Comparison with Related Work.** In addition to the adjacency matrix query model, Adriaens & Apers (2023) studied another query model based on bounded-degree graphs [5]. In this model, the adjacency list cannot directly query neighbors. Instead, they only allow queries in the form of tuples $(u, i)$, where $i$ is an integer in $[n]$. The answer is the $i$-th neighbor if $i \leq \deg(u)$, or $\perp$ otherwise. The queries in that model are inherently harder, and their query bounds are $\widetilde{O}(\sqrt{n}/\text{poly}(\varepsilon))$. While there are interesting applications in the bounded-degree model, these results are not directly comparable to ours.

The bulk of the literature in structural balance and correlation clustering has focused on computing the *clustering*, i.e., the partition of vertices. To this end, there are several popular techniques, including linear programming (Chawla et al. (2015); Cohen-Addad et al. (2022; 2023); Cao et al. (2024), pivot-based algorithms (Ailon et al. (2008); Makarychev & Chakrabarty (2023); Dalirrooyfard et al. (2024); Cambus et al. (2024); Dalirrooyfard et al. (2025)), and agreement decomposition (Cohen-Addad et al. (2021); Assadi & Wang (2022); Cohen-Addad et al. (2024a)). However, all of these techniques would need $\Omega(n)$ time to write down the formulation or the solution, which is much slower than our algorithms. Assadi et al. (2023) made an attempt to combine sampling and some of the above techniques to test the cost of correlation clustering with *small space*. Their algorithm can be used for our application as well with $\text{poly}(\log n/\varepsilon)$ time, which is worse than ours.

## 2 PRELIMINARIES

We introduce the definitions and standard techniques related to the results in this section.

**Notation.** We use $G = (V, E)$ to denote a graph, where $V$ is the set of $n$ vertices and $E$ is a set of $m$ edges. We focus on a labeled complete graph, defined below.

**Definition 1** (Labeled Complete Graphs). We say $G = (V, E^+ \cup E^-)$ is a labeled complete graph if there exists exactly one edge between each vertex pair $(u, v)$, with a label of either $(+)$ or $(-)$.

We assume access to *labeled adjacency matrix* of the graph, defined as follows.

**Definition 2** (Labeled Adjacency Matrix). We say a matrix $\boldsymbol{A} \in \{-1, 1\}^{n \times n}$ is a labeled adjacency matrix of a $n$-vertex labeled complete graph $G = (V, E^+ \cup E^-)$ where $\boldsymbol{A}_{u,v} = 1$ if $(u, v)$ is a $(+)$ edge; and $\boldsymbol{A}_{u,v} = -1$ if $(u, v)$ is a $(-)$ edge.

We assume we could query any entry of the adjacency matrix in $O(1)$ time. In particular, this also allows us to query neighbors, sample edges, and sample triangles in $O(1)$ time.

**Correlation clustering and structural balance.** On a complete labeled graph, we are able to define the problem of *(min-disagreement) correlation clustering* and *structural balance* as follows.

**Definition 3** (Correlation Clustering). Let $G = (V, E^+ \cup E^-)$ be a labeled complete graph and let $\mathcal{C} = (C_1, C_2, \cdots)$ be a clustering, i.e., partition of the vertices of $V$ into disjoint vertex sets. We define the cost of correlation clustering, $\text{cost}(G, \mathcal{C})$, as the summation of the number of $(+)$ edges crossing different clusters and the number of $(-)$ edges in the same clusters:

$$\text{cost}(G, \mathcal{C}) := \left| \left\{ (u, v) \in E^+ \mid u \in C_i, v \in C_j, i \neq j \right\} \right| + \left| \left\{ (u, v) \in E^- \mid u, v \in C_i \text{ for some } i \right\} \right|.$$

We say that $G$ is *(perfectly) clusterable* if and only if there exists an optimal clustering $\mathcal{C}^*$ that induces 0 cost. Besides, we say $G$ is *(perfectly) $k$-clusterable* if and only if there exists an optimal clustering $\mathcal{C}^* = (C_1, \ldots, C_k)$ of *exactly* $k$ (possibly empty) clusters and induces 0 cost. (In other words, $\mathcal{C}^*$ has at most $k$ non-empty clusters.) Note that a clusterable graph does *not* restrict the

---

[4]https://anonymous.4open.science/r/Correlation-Clustering-Property-Testing-3EC0/

[5]In their paper, the adjacency matrix query model is called the "dense graph model" and the adjacency list query model is called "bounded degree model".

number of clusters, i.e., we can use any number of clusters to minimize the cost. In contrast, a $k$-clusterable graph must have a perfect clustering with $\leq k$ clusters.

Structural balance is a special case of correlation clustering where $k = 2$. More formally, we say that $G$ is *(perfectly) balanced* if and only if there exists a perfect optimal clustering with 2 clusters that induces 0 correlation clustering cost.

**Property Testing for Graphs Close and Far from being Clusterable (Balanced).** Similar to typical algorithms in property testing, we allow some "slackness" between the cases: in typical property testing, the algorithm should return "YES" when the property holds, and "NO" if there is a *sufficient degree of violations* to the property. In between the cases, the algorithm is typically allowed to return anything. To the above end, we introduce the notion of the *distance* for a graph from being balanced.

**Definition 4** ($\varepsilon$-far/close from being clusterable ($k$-clusterable/balanced)). Let $G = (V, E^+ \cup E^-)$ be a labeled complete graph. We say that $G$ is at least $\varepsilon$-far (resp., at most $\varepsilon$-close) from being clusterable if we have to flip the labels of at least $\varepsilon \cdot \binom{n}{2}$ (resp., at most $\varepsilon \cdot \binom{n}{2}$) edges to make the graph clusterable.

In our work, the above definition of distance also apply to the other two distinct properties: $k$-clusterability and structural balance.

With the above terminologies, a graph is at most 0-far from being clusterable if and only if it is clusterable. For a graph that is $\varepsilon$-far from being clusterable (resp., balanced), we also call an edge $e = (u, v)$ a *false edge* if $e$ needs to be flipped in the solution that flips the minimum number of edges to make the graph cluterable (resp., balanced). We also define a *bad triangle* as a triangle that contains two $(+)$ edges and one $(-)$ edge.

We will also use the following standard form of Chernoff bound in our proofs.

**Proposition 2.1** (Chernoff bound; c.f. Alon & Spencer (2016)). *Let $X_1, X_2, \ldots, X_n$ be independent random variables such that $X_i \in [0, 1]$. Let $X = \sum_{i=1}^n X_i$. Then, for every $\delta > 0$,*

$$\Pr[|X - \mathbb{E}[X]| \geq \delta \cdot \mathbb{E}[X]] \leq 2 \cdot \exp\left(-\frac{\delta^2}{2 + \delta} \cdot \mathbb{E}[X]\right)$$

We will also use Janson's inequality Janson et al. (2011) from random graph analysis, to prove our Theorem 1. Please refer to Appendix D for its formal statement and a brief explanation.

## 3 UPPER BOUND FOR TESTING $k$-CLUSTERABILITY

We showcase our algorithm results by presenting the upper bound for testing $k$-clusterability, therefore proving Theorem 2.

Our algorithm for $k$-clusterability utilizes the algorithm for testing clusterability in Appendix E in a black-box way; and is self-contained. In fact, our proof implies that given *any* algorithm for testing clusterability running in $t(\varepsilon) \geq 1/\sqrt{\varepsilon}$ time, there is an algorithm for testing $k$-clusterability running in $O(t(\varepsilon^2))$ time, for every constant $k$.

**Theorem 2.** *Fix $k \geq 2$ and $\varepsilon \in (0, 1)$. There exists a randomized algorithm that given a labeled complete graph $G = (V, E^+ \cup E^-)$ and a parameter $\varepsilon$ answers the following*

- *If $G$ is $k$-clusterable, the algorithm always answers "YES";*

- *If $G$ is at least $\varepsilon$-far from being $k$-clusterable, the algorithm answers "NO" with probability $\geq 0.9$;*

- *In addition, if $G$ is $\left(\frac{\varepsilon^4}{10^{26} k^4 \ln^4 k}\right)$-close-to-$k$-clusterable, the algorithm answers "YES" with probability $\geq 0.9$.*

*The algorithm queries at most $O\left(\frac{k^4 \ln^4 k}{\varepsilon^4}\right)$ edges of $G$ and runs in $O\left(\frac{k^4 \ln^4 k}{\varepsilon^4}\right)$ time.*

Our algorithm is a combination of two one-sided-error algorithms, one for testing whether the graph is clusterable (Algorithm 3), another for testing whether a close-to-clusterable graph is $k$-clusterable.

When the input graph is $\varepsilon$-far from $k$-clusterable, at least one of the two algorithms will output "NO" with high probability.

We start by introducing and analyzing the second algorithm under the assumption that the input graph is clusterable. Then we show that the algorithm also works for graphs that are close enough to be clusterable.

---

**Algorithm 1. An algorithm that distinguishes clusterable graphs from $k$-clusterable graphs**
**Input:** A labeled complete graph $G = (V, E^+ \cup E^-)$ that is clusterable;[a]

1. Sample a subset $S$ of $s = \min\left(100\frac{k\ln k}{\varepsilon}, n\right)$ vertices from $V$ uniformly at random (with replacement).
2. Maintain $k$ subsets $S_1, \ldots, S_k$ of $S$. Initially, all the sets are empty.
3. For each vertex $u \in S$ and each $i \in \{1, \ldots, k\}$, query if $(u, v) \in E^+$ where $v$ is an arbitrary vertex from $S_i$.
4. For the first time when a positive edge is discovered between $u$ and a vertex $v$ in $S_i$, add $u$ to $S_i$.
5. If $u$ has no positive edge to any of the subsets $S_i$, add $u$ to an empty subset $S_j$. In addition, if there is no empty subset that $u$ can add to, return "NO".
6. After iterating all the vertices in $S$, return "YES".

---

[a]We assume for now that $G$ is clusterable. And we will show that with high probability a random subgraph of $G$ will still be clusterable when $G$ is close-enough-to-clusterable; and the algorithm cannot distinguish the two cases. See Lemma 3.2 for details.

---

**Lemma 3.1.** *Fix parameters $k \geq 2$ and $\varepsilon \in (0, 1)$. Given a labeled complete graph $G = (V, E^+ \cup E^-)$, Algorithm 1 answers as follows*

- *If $G$ is $k$-clusterable, the algorithm always answers "YES";*

- *If $G$ is clusterable but is at least $\varepsilon$-far from being $k$-clusterable, the algorithm answers "NO" with probability $\geq 9/10$;*

- *In addition, if $G$ is $\left(\frac{\varepsilon^2}{10^6 k^2 \ln^2 k}\right)$-close-to-$k$-clusterable, the algorithm answers "YES" with probability $\geq 99/100$.*

*Besides, Algorithm 1 queries at most $O(\frac{k^2 \ln k}{\varepsilon})$ edges of $G$ and runs in $\tilde{O}(\frac{k^2 \ln k}{\varepsilon})$ time.*

We defer the proof to Lemma 3.1 to Appendix C. Our analysis relies on the fact that the input graph is clusterable. However, we will show that when the input graph is $\left(\delta := \frac{\varepsilon^2}{10^6 k^2 \ln^2 k}\right)$-close to clusterable but $\varepsilon$-far from $k$-clusterable, the above algorithm still works with high probability. Intuitively, when the input graph is guaranteed to be $\delta$-close to clusterable, a random $\Theta(\frac{k\ln k}{\varepsilon})$-size subgraph will not contain any false edge with high probability. This observation is formalized as the following lemma.

**Lemma 3.2.** *Fix $\delta \in (0, 1)$. Given labeled complete graphs $G = (V, E^+ \cup E^-)$ and $G' = (V, E^+ \cup E^-)$ such that $G'$ is obtained by flipping at most $\delta\binom{n}{2}$ edges from $G$. Let $S$ a subset of $s$ vertices from $V$ selected uniformly at random (with replacement). Let $G_S, G'_S$ denote the induced subgraph by the sampled vertices. If $s \leq \frac{1}{10\sqrt{\delta}}$,*

$$\Pr_S[G_S = G'_S] \geq 99/100.$$

*Proof.* We show by union bound that with $\geq 99/100$ probability the sampled subgraph does not contain any edge in $G - G'$, the edges with labels flipped in $G$ to obtain $G'$.

For every single edge in $G - G'$, this edge is sampled with probability at most

$$\sum_{1 \leq i < j \leq s} 2 \cdot \frac{1}{n^2} \leq s^2/n^2.$$

where the factor of 2 counts for the same edge $(u, v)$ of different orders ($u$ is sampled at the $i$-th place, $v$ is sampled at the $j$-th place; or vice versa). Summing over all the $\leq \delta \binom{n}{2}$ edges in $G - G'$, the probability that any of the edges from $G - G'$ is sampled is at most $delta \binom{n}{2} \cdot \frac{s^2}{n^2} \leq 1/100$.

$\square$

Now we formally give and analyze the algorithm that combines Algorithm 3 and Algorithm 1.

---

**Algorithm 2. An algorithm for testing $k$-clusterability**
**Input:** A labeled complete graph $G = (V, E^+ \cup E^-)$; a parameter $\varepsilon$.

1. Independently run Algorithm 3 with parameter $\delta := \frac{\varepsilon^2}{10^6 k^2 \ln^2 k}$ twice. If Algorithm 3 ever answers "NO", return "NO";
2. Independently run Algorithm 1 with parameter $\varepsilon/2$ twice. If Algorithm 1 ever answers "NO", return "NO";
3. If all the above simulations answer "YES", return "YES".

---

*Proof to Theorem 2.* We claim that Algorithm 2 is the desired algorithm that distinguishes whether a graph is $k$-clusterable or $\varepsilon$-far from $k$-clusterable.

The query complexity and the time complexity of Algorithm 2 are dominated by calling Algorithm 3 twice, which costs $O(\frac{k^4 \ln^4 k}{\varepsilon^4})$ queries and time.

Given a graph that is $k$-clusterable, by Theorem 1 and Lemma 3.1, the algorithm will always answer "YES". In addition, by Lemma 3.2, for a graph $G$ that is $\left( \frac{\varepsilon^4}{10^{26} k^4 \ln^4 k} \right)$-close-to-$k$-clusterable, Algorithm 1 (sampled $200 \frac{k \ln k}{\varepsilon}$ vertices) and Algorithm 3 (sampled $\frac{10^{12} k^2 \ln^2 k}{\varepsilon^2}$ vertices) will return "YES" with probability $\geq 99/100$. By a union bound, the final output is "YES" with probability $\geq 0.9$.

If the graph is $\delta$-far from clusterable, Theorem 1 guarantees that the answer will be "NO" with $\geq 9/10$ probability. The remaining case is when the input graph $G$ is $\delta$-close-to-clusterable but $\varepsilon$-far-from-$k$-clusterable. Assume $G'$ to be the clusterable graph obtained by flipping at most $\delta \cdot \binom{n}{2}$ edges of $G$. By Lemma 3.2, Algorithm 1 returns the testing answer of $G'$ with $\geq 99/100$ probability. By Lemma 3.1, Algorithm 1 returns "NO" with probability $\geq 89/100$ given $G$.

In this case, the probability that all the tests fail is at most $\leq 0.11^2 < 0.1$. Therefore, Algorithm 2 outputs "NO" with $\geq 9/10$ probability when the input is $\varepsilon$-far from being $k$-clusterable. $\square$

## 4 EXPERIMENTS

We assess the empirical performance of testing correlation clustering with three proposed algorithms: Algorithm 3 for clustering with general $k$, Algorithm 2 for clustering with fixed $k$, and Algorithm 4 for structural balance. The evaluation metrics include query complexity, running time and testing accuracy in practice. There exists only one baseline from prior work Adriaens & Apers (2023), where the tester for structural balance in the adjacency matrix query model is implemented.

**Setup.** Since the CC problem is NP-hard, obtaining the ground-truth $\varepsilon$-farness becomes a challenge. To address this, we generate **synthetic graphs** based on 6 different perturbation schemes to the well-clustered signed graph such that the optimal cost and the number of clusters are tractable. We explain them in Appendix B. Together with the balanced/0-cost case, we use these synthetic graphs of 7 scenarios (in total 140 instances) for experiments. Some basic statistics are shown in Table 2. For structural balance experiments, we set $n = 1000$ and $k = 2$. To facilitate testing on **real-world** graphs, we use the spectral frustration index to obtain an approximation of the ground truth $\varepsilon$-farness with respect to testing structural balance. We also demonstrate the spectrum of testing outcome as $\varepsilon$ increases from 0.05 to 0.5, for both structural balance and general CC testing.

Table 2: Synthetic Signed Graphs and Ground Truth $\varepsilon$ used in CC testing experiments

| Model | Pure | Uniform-noise | Hetero-noise | Cycle | Half-flip | Cluster-swap | Mixed-flip | size $n$ | $k$ |
|---|---|---|---|---|---|---|---|---|---|
| $\varepsilon$ Range | 0 | $0.32 \sim 0.49$ | $0.28 \sim 0.42$ | 0.30 | $0.30 \sim 0.38$ | 0.25 | 0.4 | 5000 | 5 |

## 4.1 TESTING ON SYNTHETIC GRAPHS

With ground truth $\varepsilon$, we are able to report the testing accuracy for synthetic graphs. All of our testers are one-sided, therefore the accuracy is defined as the percentage of the correct output of "YES/No" corresponding to the label of balance or not. Note that our algorithms use large constants for the convenience of proof, in practice we only make it at most 3 unless mentioned otherwise.

Table 3: Testing Performance with $\varepsilon = 0.1$

| Algorithm | Accuracy | Query Complexity (# sampled edges) | Running Time (s/graph) |
|---|---|---|---|
| Test CC (general $k$), Algorithm 3 | 1.0 | 10000 | 23.8 |
| Test CC (fixed $k$)[6], Algorithm 1 | 1.0 | 1610 | 22.5 |
| Test Structural Balance, Algorithm 4 | 1.0 | 60 | $1.3 \times 10^{-4}$ |
| Test Structural Balance, Adriaens & Apers (2023) | 1.0 | 900 | 1.1 |

Table 3 shows that our algorithms for testing CC and structural balance yield favorable efficiency on query complexity and running time. For testing structural balance, comparing to Adriaens & Apers (2023), our algorithm requires significantly smaller sampling size and runtime. For testing CC with fixed $k$, we collect results for $k = 3, 4, 5$ on "pure" model graphs. Finally, all algorithms give testing accuracy 1, showing the effectiveness of the algorithms, thus corroborate with the theoretical results.

We next demonstrate the performance on the same set of metrics as $\varepsilon$ increases from 0.05 to 0.5 for structural balance in Figure 1. Two algorithms are similar on accuracy, which has small fluctuations but remains higher than 0.95. But for efficiency, we observe that Algorithm 4 outperforms the baseline algorithm by a large margin, especially when $\varepsilon$ is small.

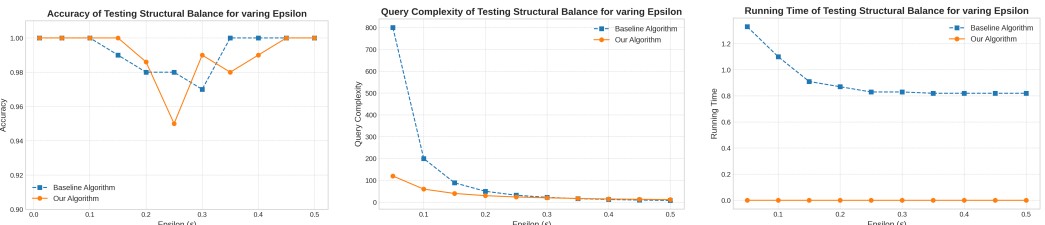

Figure 1: Performance on structural balance testing with varying $\varepsilon$.

**Scalability.** The theoretical results show that the query complexity does not involve $n$, the size of graph. Therefore it is conceivable that the algorithms are scalable. We examine this issue in practice, by showcasing the performance of Algorithm 3 for testing CC as $n$ scales up to 50000. Table 4 demonstrates that the testing algorithm itself still executes efficiently, however the entire program involving processes such as sampling, may become prohibitive in practice as $n$ scales.

Table 4: Running Time of CC testing with $\varepsilon = 0.1$

| Graph size | 10000 | 20000 | 30000 | 40000 | 50000 |
|---|---|---|---|---|---|
| Testing Algorithm Runtime | 0.011 | 0.013 | 0.015 | 0.17 | 0.20 |
| Total Runtime (log) | 4.51 | 6.36 | 7.81 | 9.89 | 12.03 |

## 4.2 TESTING ON REAL-WORLD GRAPHS

We move forward to evaluating Algorithm 3 and Algorithm 4 on 6 real-world graphs selected from the SNAP project[7]. The datasets encompass social, financial, collaboration and communication networks, with varying sizes between 500 and 10000. In the experiments with real-world graphs,

---

[6]For practicability we test the scenario: the input graph is clusterable but not clear if $k$-clusterable
[7]https://snap.stanford.edu/data/

we treat the edges in the graphs as $(+)$ edges and the non-edge vertex pairs as $(-)$ edges. The reduction suits well for our datasets, where the $(-)$ relationships (e.g., no message exchanges) can be directly inferred from the $(+)$ relationships (e.g., has message exchanges). We illustrate the spectrum of the testing output for both tasks, as $\varepsilon$ increases from 0.05 to 0.5 in Figure 2. Although the labels are missing, we are able to approximate the structural balance frustration index using the smallest eigenvalue of the signed Laplacian matrix Kunegis et al. (2010). We then obtain an $\varepsilon'$ as a lower bound of the ground truth $\varepsilon$, which is a signal of the testing correctness.

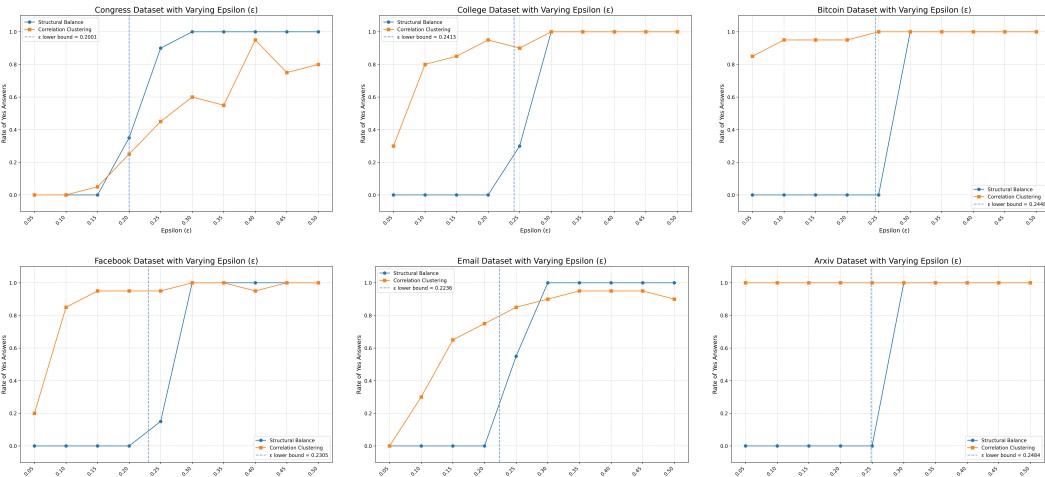

Figure 2: Testing output for all real-world graphs. Results are averaged on 20 repeated runs. The dotted light blue vertical line shows the lower bound of the true $\varepsilon$.

First, we observe from Figure 2 that all testing results transits from "NO" to "YES". Structural balance has a clearer phase transition structure than correlation clustering, and the transition happens right after the estimated $\varepsilon$ lower bound, which is supporting evidence of the testing accuracy. In other words, initially, both algorithms report "not balanced" (resp. "not clusterable") due to the fact that $\varepsilon$ value is very small, and the condition to pass the test is very stringent. As we increase the value $\varepsilon$, the algorithm demonstrates a "tolerate test" property such that it allows the graph to be report as "balanced" (resp. "clusterable") when the graphs are relatively close to being balanced (resp. clusterable) with the given $\varepsilon$ parameter.

All experiments take a very short time ($< 0.1s$), showing the potential of our algorithms in real-world applications. Another interesting observation is that many (in our case, all) real-world graphs have $\varepsilon$-farness with $\varepsilon \leq 0.3$.

## 5 DISCUSSION AND CONCLUSION

Our analysis of using Janson's inequality may also be of independent interest. It outperforms the classic analysis using the graph removal lemma in the labeled graphs, and provides a more fine-grained way of analyzing subgraph testing algorithms. Our proof technique may find broader applications in analyzing property testing algorithms.

Another future direction would be to generalize our results to *general labeled graphs*, where only a subset of all $\binom{n}{2}$ edges are labeled. This setting is more aligned with real-world applications, but it poses a significant challenge: our algorithms fundamentally rely on detecting local patterns, such as inconsistent triangles. In a sparse graph, a graph that is globally far from being clusterable may not contain any such local witnesses, rendering our current approach ineffective.

## ACKNOWLEDGEMENT

The authors would like to thank Vikrant Ashvinkumar for insightful discussions and anonymous ICLR reviewers for valuable feedback. Deng and Gao would like to acknowledge funding support from NSF through CCF-2118953, DMS-2311064, DMS-2220271, IIS-2229876, and CNS-2515159.

## USAGE OF GENAI

We use GenAI to check typos and writings in the paper.

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

## A  MORE DISCUSSIONS ON RELATED WORK

As discussed, the bulk of the literature in structural balance and correlation clustering has focused on computing the *clustering*, i.e., the partition of vertices. To this end, there are several popular techniques, including linear programming (Chawla et al. (2015); Cohen-Addad et al. (2022; 2023); Cao et al. (2024), pivot-based algorithms (Ailon et al. (2008); Makarychev & Chakrabarty (2023); Dalirrooyfard et al. (2024); Cambus et al. (2024); Dalirrooyfard et al. (2025)), and agreement decomposition (Cohen-Addad et al. (2021); Assadi & Wang (2022); Cohen-Addad et al. (2024a)). However, all of these technique would need $\Omega(n)$ time to write down the formulation or the solution, which is much slower than our algorithms. Assadi et al. (2023) made an attempt to combine sampling and some of the above techniques to test the cost of correlation clustering with *small space*. Their algorithm can be used for our application as well with $\text{poly}(\log n/\varepsilon)$ time, which is worse than ours.

The problem of testing whether a graph is clusterable (resp. balanced) is related to the MAX-CSP formulation. In the generic $r$-MAX-CSP problem, we are given $m$ boolean functions, and each of the function uses at most $r$ variables. Alon et al. (2003) provided a generic framework that approximates the number of satisfiable functions by querying $O(\frac{\log 1/\varepsilon}{\varepsilon^4})$ variables. In the problem of testing structural balance and clusterability, we define a Boolean variable for each vertex, and for each edge $e = (u, v) \in E$ we define a function $f_e$ that encodes the "right assignment" of the vertex variables with respect to the label of the edge: $f$ is satisfied if $(u, v) \in E^+$ and $u, v$ are in the same cluster or $(u, v) \in E^-$ and $u, v$ are in different clusters. It is easy to see for this application, we have $r = 2$, which induces additive error of $\varepsilon n^2$: this satisfies the definition of $\varepsilon$-far from being balanced and/or clusterable. However, such a strategy leads to the algorithm in Adriaens & Apers (2023), which gives suboptimal bounds.

## B MORE DETAILS ON EXPERIMENTS

All of our experiments are implemented with Intel Core i9 CPU of 32GB memory, no GPU is required. Now we introduce the synthetic graph generation models. For the "good" case, the generation is straightforward: create $k$ clusters first, put every edge inside each cluster with positive sign and vice versa. Below shows the perturbation for graphs in the potential "bad" case, namely the optimal cost is large.

- **Uniform Noise:** Each sign is flipped with a uniform probability $p \in [0.3, 0.5]$.

- **Heterogeneous Noise:** The signs of intra-cluster edges (+1) are flipped with probability $p_{in} \in [0.2, 0.4]$, while the signs of inter-cluster edges (-1) are flipped with probability $p_{out} \in [0.3, 0.5]$.

- **Cycle:** The $k$ clusters are arranged in a cycle. Ideal edges are set to +1 if they are within a cluster or between adjacent clusters in the cycle, and -1 otherwise. All edge signs are then flipped with a 30% probability.

- **Half Flip:** One cluster is chosen at random. The sign of every edge incident to this chosen cluster is then flipped with a 50% probability.

- **Cluster Swap:** One cluster is chosen at random, for half of its nodes, the signs of all edges connecting them to any node outside the original cluster are flipped.

- **Mixed Flip:** flip 40% edge signs inside each cluster, and 40% across clusters

**Signed Laplacian and spectral approximation of frustration.** Denote the *frustration index* as $f(G)$. Let $W \in \mathbb{R}^{n \times n}$ denote the signed adjacency matrix of a graph on $n$ nodes, where $w_{ij} \in \{-1, 0, +1\}$ (or more generally real weights, but not in our context). Define the *absolute degree* $d_i = \sum_j |w_{ij}|$, and let $D = \mathrm{diag}(d_1, \ldots, d_n)$. The *signed Laplacian* is

$$L = D - W.$$

For any assignment $x \in \{\pm 1\}^n$ one has the identity

$$x^\top L x = \sum_{i<j} |w_{ij}| (x_i - \mathrm{sign}(w_{ij}) x_j)^2.$$

When $w_{ij} \in \{\pm 1\}$ this reduces to

$$x^\top L x = 4 \cdot \big(\# \text{ of frustrated edges under assignment } x\big).$$

Thus minimizing $x^\top L x$ over $\{\pm 1\}^n$ is equivalent to computing $f(G)$. By the Rayleigh–Ritz principle,

$$\lambda_{\min}(L) = \min_{y \neq 0} \frac{y^\top L y}{y^\top y}.$$

For any $\{\pm 1\}$ vector $x$, since $\|x\|^2 = n$, we obtain

$$\frac{x^\top L x}{x^\top x} = \frac{4f(x)}{n},$$

where $f(x)$ is the number of frustrated edges under $x$. Minimizing over all $x$ yields

$$\lambda_{\min}(L) \leq \frac{4f(G)}{n}.$$

Equivalently,

$$f(G) \geq \frac{n}{4} \lambda_{\min}(L).$$

Hence the scaled smallest eigenvalue $\frac{n}{4}\lambda_{\min}(L)$ provides a computable spectral lower bound on the frustration index. This is known as the *spectral approximation of the frustration index*, and it has been used as a tractable proxy for quantifying balance in signed networks Kunegis et al. (2010).

## C  MISSING PROOF TO LEMMA 3.1

*Proof to Lemma 3.1.* The time and query complexity of Algorithm 1 is clear. It samples $O(\frac{k \ln k}{\varepsilon})$ vertices. For each iterated vertex $u$, it will perform at most $k$ queries. The total number of queries is $O(\frac{k^2 \ln k}{\varepsilon})$. In addition, maintaining the subsets costs time $\tilde{O}(\frac{k^2 \ln k}{\varepsilon})$.

Given a $k$-clusterable graph $G$. Any induced subgraph of $G$ can be partitioned into at most $k$ clusters. Algorithm 1 will always output "YES". Given a $\left(\frac{\varepsilon^2}{10^6 k^2 \ln^2 k}\right)$-close-to-$k$ clusterable graph, by Lemma 3.2, Algorithm 1 will output "YES" with probability $\geq 99/100$.

Given a graph $G$ that is clusterable but $\varepsilon$-far from $k$-clusterable. Let $t$ be the number of clusters in $G$. $G$ can be characterized by a list of cluster sizes $(s_1, \ldots, s_t)$ where $\sum_{i=1}^t s_i = n$. Without loss of generality, we assume

$$s_1 \geq s_2 \geq \cdots \geq s_t.$$

In addition, we may assume that $t > k$, since otherwise a graph of $t$ clusters is also $k$-clusterable, by appending $k - t$ empty clusters.

Let $r = n - \sum_{i=1}^k s_i$. Then $r \geq \varepsilon n/2$, or otherwise we can make $G$ a $k$-clusterable graph by merging the last $t - k$ clusters into the first cluster, which costs $< r(n - r) < \varepsilon \binom{n}{2}$ flips.

When $s_k \geq \frac{\varepsilon n}{20k}$, the largest $k$ clusters are large enough, and the subset $S$ contains vertices from all the largest $k$ clusters and a vertex from the last $t - k$ clusters with high probability. Formally, by union bound, the probability that any of the first $k$ clusters or the union of the last $r$ vertices does not have vertices in $S$ is at most

$$\sum_{i=1}^k \left(1 - \frac{\varepsilon}{20k}\right)^s + (1-\varepsilon)^s \leq \sum_{i=1}^{k+1} \left(1 - \frac{\varepsilon}{20k}\right)^s \leq \sum_{i=1}^{k+1} \exp\left(-\frac{\varepsilon s}{20k}\right) \leq (k+1) \exp\left(-5 \ln k\right) \leq 1/10.$$

In this case, with $\geq 9/10$ probability the sampled subset contains an independent set (i.e., with no positive edges in between) of size $\geq k + 1$ and the algorithm outputs "NO".

When $s_k < \frac{\varepsilon n}{20k}$, we know $s_j \leq s_k < \frac{\varepsilon n}{20k}$ for any $j \geq k$ since the clusters are sorted in decreasing order of size. In this case we call all clusters but the largest $k$ as small clusters. Since $r \geq \varepsilon n/2$, the number of small clusters is at least $\frac{r}{s_k} \geq 10k$. We show that with high probability the sample set $S$ includes vertices from $\geq k + 1$ different small clusters.

Let $X_1, \ldots, X_s \in \{0, 1\}$ be random bits indicating whether each sample covers a small cluster that is never sampled in its previous samples, and let $X = \sum_{i=1}^s X_i$ be their sum. For every $i \in \{1, \ldots, s\}$, we have

$$\Pr\left[X_i = 1 \middle| \sum_{j=1}^{i-1} X_i \leq k\right] \geq \frac{r - k \cdot \frac{\varepsilon n}{10k}}{n} \geq 0.4\varepsilon.$$

Let $Y_1, \ldots, Y_s \in \{0, 1\}$ be independent random bits where $\Pr[Y_i = 1] = 0.4\varepsilon$ for each $Y_i$, and $Y$ their sum. Then the sum of $(X_i)$ is dominated by the sum of $(Y_i)$.[8] Note that $\mathbb{E}[Y] = 0.4\varepsilon s = 40k \ln k$. By applying the Chernoff bound (Proposition 2.1) and setting $\delta = 1 - \frac{1}{40 \ln k}$, we have

$$\Pr[X \leq k] \leq \Pr[Y \leq k] \leq 2 \cdot \exp\left(-\frac{\delta^2}{2+\delta} \cdot \mathbb{E}[Y]\right) < 2 \cdot \exp(-12k \ln k) < 1/10.$$

To conclude, in both cases, Algorithm 1 can sample an independent set of size $\geq k + 1$ and output "NO" with high probability. □

## D  PRELIMINARIES ON JANSON'S INEQUALITY

In this section, we briefly review Janson's inequality, a fundamental tool from the probabilistic method. We employ this inequality in our analysis (Lemma E.2 and the proof of Theorem 1) to

---

[8]This can also be shown by a standard coupling argument.

bound the probability that a sum of dependent yet structured indicator random variables equals zero. While the standard Chernoff bound applies to sums of *independent* random variables, Janson's inequality provides strong bounds for sums of variables that exhibit local dependencies.

**Lemma D.1.** *(Janson's inequality; c.f. Janson et al. (2011)) Let $n \geq 1$ be an integer. Let $\Gamma$ be a random subset of $[n]$ such that for each $i \in [n]$, $i \in \Gamma$ with independent probability $p_i$.*

*Let $R$ be a family of subsets of $[n]$. For every $A \in R$, let $I_A$ be the indicator random variable such that $I_A = 1$ if and only if $A \subseteq \Gamma$, and $I_A = 0$ otherwise. Let $X$ be the random variable denoting the number of sets in $R$ that are subsets of $\Gamma$. Then*

$$\Pr[X = 0] \leq \exp\left( \min\left( -\lambda + \Delta, -\frac{\lambda^2}{\lambda + 2\Delta} \right) \right),$$

*where $\lambda = \mathbb{E}[X]$ and $\Delta = \frac{1}{2} \sum_{A,B \in R : A \neq B, A \cap B \neq \emptyset} \mathbb{E}[I_A I_B]$.*

**Relevance to our analysis.** In our proofs, we frequently search for a "witness" structure (such as a negative edge connected by a positive path) within the induced subgraph of sampled vertices. Since multiple potential witnesses may share vertices, their appearances are not independent. Janson's inequality allows us to lower-bound the probability of finding at least one such witness by controlling the overlapping term $\Delta$. Specifically, when $\Delta$ is small relative to $\lambda$, the bound behaves similarly to the Chernoff bound ($\approx e^{-\lambda}$); when correlations are high ($\Delta > \lambda$), the probability decays as $\approx e^{-\lambda^2/2\Delta}$.

## E  A DIRECT ALGORITHM FOR TESTING CLUSTERABILITY

We present the algorithm for testing general clusterability using $O(1/\varepsilon^2)$ time and queries in this section. We first recall the statement of the result.

**Theorem 1.** *Fix $\varepsilon \in (0, 1)$. There exists a randomized algorithm that given a labeled complete graph $G = (V, E^+ \cup E^-)$ and a parameter $\varepsilon$ answers with the following rules*

- *If $G$ is clusterable, the algorithm always answers "YES";*

- *If $G$ is at least $\varepsilon$-far from being clusterable, the algorithm answers "NO" with probability $\geq 0.9$;*

- *If $G$ is $C \cdot \varepsilon^2$-close to being clusterable for some small constant $C$, the algorithm answers "YES" with probability $\geq 0.9$.*

*The algorithm queries at most $O(1/\varepsilon^2)$ edges of $G$ and runs in $O(1/\varepsilon^2)$ time.*

While standard combinatorial arguments often use the graph removal lemma, a direct application of Fox's colored graph removal lemma only yields an upper bound of $\tilde{O}(\text{tower}(\log(1/\varepsilon)))$ for testing bad-triangle-freeness Ruzsa & Szemerédi (1978); Fox (2011); Adriaens & Apers (2023).[9] Besides, a reduction to the MAX-CSP problem also gives a two-sided error algorithm of $\tilde{O}(1/\varepsilon^7)$ query complexity and $\exp(\tilde{O}(1/\varepsilon^3))$ running time Andersson & Engebretsen (2002); Adriaens & Apers (2023).

We overcome this limitation by employing Janson's inequality, a classic tool from random graph theory, to constructively demonstrate the existence of *bad triangles* in a small sample, which we will define later. To the best of our knowledge, this is also the first time Janson's inequality is used in analyzing property testing algorithms. Our proof may be of independent interest. Compared to the algorithm in Adriaens & Apers (2023), our work provides a one-sided error algorithm, drastically improving both the query complexity and the running time to $O(1/\varepsilon^2)$.

Our algorithm is simple: we sample $O(1/\varepsilon)$ vertices, query their induced subgraph, and check whether there is any inconsistency.

---

[9]The towering function $\text{tower}(x)$ denotes a tower of 2's of height $x$, i.e., 2-to-the-2-to-the-...-to-the-2, $x$ times. Thus, $\tilde{O}(\text{tower}(\log(1/\varepsilon)))$ is much larger than $1/\text{poly}(\varepsilon)$. In fact, $\text{tower}(6)$ is more than the estimated number of elementary particles in the observable universe.

---

**Algorithm 3. An algorithm for testing clusterability**
**Input:** A labeled complete graph $G = (V, E^+ \cup E^-)$; a parameter $\varepsilon$.

1. Sample a subset $S$ of $s = \min(10^6/\varepsilon, n)$ vertices from $V$ uniformly at random (without replacement).
2. Let $G_S$ be the complete subgraph induced by the sampled vertices.
3. Run breadth-first search (BFS) to check whether $G_S$ contains bad triangles, i.e., a triangle $(u, v), (v, w), (u, w)$ among which exactly two edges are $(+)$ and one edge is $(-)$. If $G_S$ contains no bad triangle, return "YES". Otherwise, return "NO".

---

Specifically, we run BFS on the positive edges of the subgraph $G_S$ and check if they form a set of vertex-disjoint complete subgraphs, which costs $O(1/\varepsilon^2)$ time.

For clusterable graphs, by proposition E.1, the algorithm will always output "YES". We show that for every graph that is $\varepsilon$-far from being clusterable, the algorithm will output "NO" with $\geq 9/10$ probability. In addition, instead of showing that the algorithm can find bad triangles with high probability, we analyze a similar pattern of subgraphs, which will simplify our analysis.

**Proposition E.1.** *Given a complete labeled graph $G = (V, E^+ \cup E^-)$. The following three conditions are equivalent.*

1. *$G$ is clusterable;*

2. *$G$ does not contain any bad triangle.*

3. *There does not exist an edge $(u, v) \in E^-$ that is connected in $G' = (V, E^+)$.*

We call the path in $G'$ that connects $u$ and $v$ a *positive path between $u$ and $v$*.

*Proof.* Let us first show that $G$ is clusterable if and only if it does not contain any bad triangle. Given a clusterable graph $G$, by the definition of clusterable graphs, there exists a clustering $\mathcal{C}$ of $G$ with a cost of 0. If there is a bad triangle $(u, v), (v, w) \in E^+$ and $(u, w) \in E^-$, $u, v$ (resp., $v, w$) must belong to the same cluster. However, $(u, w) \in E^-$ implies that $u, w$ cannot belong to the same cluster. This contradiction implies that graphs with bad triangles cannot be clusterable.

Now, we argue that graph $G$ that is not clusterable must contain at least one bad triangle. Let $S \subset V$ be a subset of vertices, and $u \notin S$ a vertex, such that the induced subgraph $G_S$ over $S$ is clusterable, but $G_{S \cup \{u\}}$ is not clusterable. By the definition, there exists a clustering $\mathcal{C} = (C_1, C_2, \dots)$ of $G_S$ with a 0 cost. We discuss three different cases. When $u$ connects to all the vertices in $S$ by $(-)$ edges, the clustering $\mathcal{C}' = (\{u\}, C_1, C_2, \dots)$ will have 0 cost, contradicting to the assumption that $G$ is not clusterable. When $u$ has $(+)$ edges only to one of the clusters (without loss of generality, we assume it is $C_1$), there must exists a vertex $w \in C_1$ such that $(u, w) \in E^-$. Otherwise $\mathcal{C}' = (\{u\} \cup C_1, C_2, \dots)$ will have 0 cost. In addition, we assume $v \in S$ is one of the vertices such that $(u, v) \in E^+$. By the fact that $v, w \in S$, $(v, w) \in E^+$. $(u, v, w)$ forms a bad triangle. Lastly, when $u$ has $(+)$ edges to multiple clusters in $\mathcal{C}$, we assume $(u, v), (u, w) \in E^+$ where $v, w$ belong to different clusters. Then $(v, w) \notin E^-$ and $(u, v, w)$ forms a bad triangle.

What is remained is to show that $G$ contains a bad triangle if and only if it contains a positive path enclosed by a negative edge.

Since a bad triangle itself is such a cycle, we only need to prove the "if" direction. Given a negative edge $(u, v)$ and a positive path $P = u \to w_1 \to \cdots \to w_t \to v$ connecting $u$ and $v$. For simplicity, we denote $w_0 = u$ and $w_{t+1} = v$. Suppose for the sake of contradiction that $G$ does not contain any bad triangle. We show by induction that for every $d \geq 2$ and every $i \in [0, t+1-d]$, $(w_i, w_{i+d}) \in E^+$, which is a contradiction to $(u, v) \in E^-$.

The base case is when $d = 2$, for every $i \in [0, t-1]$, $(w_i, w_{i+2}) \in E^+$ or otherwise $(w_i, w_{i+1}, w_{i+2})$ form a bad triangle. Suppose the above is true for every $d < d_0$. For every $i \in [0, t+1-d_0]$, $(w_i, w_{i+d-1}) \in E^+$ by our inductive hypothesis. Then, $(w_i, w_{i+d})$ must be positive or otherwise $(w_i, w_{i+d-1}, w_{i+d})$ will form a bad triangle. $\qquad\square$

Our proof of correctness discusses three different types of graphs in the "NO" case. Before we delve into the proof details, below are necessary definitions and lemmas that will be used in our proof.

**Definition 5.** Given the labeled graph $G = (V, E^+ \cup E^-)$, we let $\mathcal{C} = (C_1, C_2, \ldots, C_k)$ be an optimal correlation clustering of $G$. If there are multiple optimal clusterings, we fix an arbitrary minimal optimal clustering, i.e., for every cluster $C \in \mathcal{C}$, splitting $C$ into two non-empty clusters will always increase the clustering cost.

The clustering $\mathcal{C}$ defines an equivalence of the vertex set. We use $u \sim_{\mathcal{C}} v$ or simply $u \sim v$ to denote that $u, v \in V$ belong to the same cluster of $\mathcal{C}$. We call edges $(u, v) \in E^-$ but $u \sim v$ as *false negative edges*. And we call edges $(u, v) \in E^+$ but $u \not\sim v$ as *false positive edges*.

Denote by clusters in $\mathcal{C}$ of $\geq \varepsilon n/20$ vertices as *large clusters*, and those of size $< \varepsilon n/20$ *small clusters*. Let $F = F_N \cup F_P$ denote the set of false edges, i.e., flipping edges in $F$ will yield a clusterable graph, where $F_P$ and $F_N$ refer to the set of false positive edges and false negative edges respectively. In addition, we split $F_P$ into two disjoint subsets $F_P = F_{P,L} \cup F_{P,S}$. $F_{P,L}$ indicates the set of false positive edges $(u, v)$ where *at least one* of $u, v$ belong to large clusters. And $F_{P,S}$ indicates the set of false positive edges $(u, v)$ whose both endpoints belong to small clusters.

We will use different proof strategies to prove the correctness of our algorithm in the following three cases.

- Case 1: $|F_N| \geq |F_P|$.
- Case 2: $|F_N| < |F_P|$ and $|F_{P,L}| \geq |F_{P,S}|$.
- Case 3: $|F_N| < |F_P|$ and $|F_{P,L}| < |F_{P,S}|$.

Our proof repeatedly uses Janson's inequality, which helps us connect the number of vertices sampled and the probability of including a bad triangle in the queried subgraph. We do not directly analyze the number of bad triangles in the sampled subgraph using Janson's inequality, because we do not even know how many bad triangles are there in an arbitrary graph from the NO case. Instead, we find a negative edge and a positive path connecting it, where the existence of each vertex of such a path will be guaranteed by Janson's inequality.

Both our proofs to Case 1 and Case 2 use the following lemma as a subroutine, which is built on Janson's inequality.

**Lemma E.2.** *Let $n \geq 1$ be a large enough integer and $G = (V, E^+ \cup E^-)$ be an arbitrary labeled graph of $n$ vertices. Let $\varepsilon \in (0, 1)$ be a fixed parameter, and $c \in [0.01, 1], c' \in [0.25, 1]$ be arbitrary fixed constants. Fix a minimal optimal clustering $\mathcal{C}$ of $G$. Let $T$ be a random subset of $V$ such that each vertex is included in $T$ with independent probability $p := \min(\frac{2 \cdot 10^5}{\varepsilon n}, 1)$. Fix a cluster $C$ in $\mathcal{C}$ of size $\geq c \cdot \varepsilon n$, a vertex $u \in C$, and a set $C' \subseteq C$ such that $|C'| \geq c' \cdot |C|$. With $\geq 99/100$ probability there exists a vertex $v \in C'$ such that the induced subgraph over $T \cup \{u\}$ contains a positive path between $u$ and $v$.*

*Proof.* Our key observation is a win-win argument. Let $N_C(u)$ denote the set of neighbors of $u$ connected by positive edges in $C$. Notice that $|N_C(u)| > (|C| - 1)/2$ or otherwise splitting $u$ out from $C$ will not decrease the clustering cost, contradicting to our assumption that $\mathcal{C}$ is the minimum optimal clustering. Since $|N_C(u)|$ is an integer, equivalently $|N_C(u)| \geq |C|/2$. At a high-level, when $|N_C(u) \cap C'| = \Omega(\varepsilon n)$, with high probability the set $T$ contains a vertex in $N_C(u) \cap C'$. When $|N_C(u) \cap C'|$ is small, the number of (+) edges between $N_C(u)$ and $C'$ should still be $\Omega(\varepsilon^2 n^2)$ or otherwise splitting $C'$ out from the cluster $C$ will yield a better clustering, which contradicts the optimality of $\mathcal{C}$. Such an edge will be sampled with high probability by applying Janson's inequality. For the special case where $C'$ is the set of positive neighbors of a vertex $v \in C$, we refer to Figure 3 for an illustration of our ideas.

Specifically, when $|N_C(u) \cap C'| \geq 0.1c'|C|$, the probability that none of these vertices are sampled in $T$ is at most
$$(1 - p)^{0.1c'|C|} \leq e^{-0.1c'p|C|} < 1/100$$
Hence $T$ contains a vertex in $C'$ that is connected to $u$ by a positive edge with high probability.

Now we assume $|N_C(u) \cap C'| < 0.1c'|C|$. Then $|C' - N_C(u)| > 0.9c'|C|$. By our assumption that $\mathcal{C}$ is a minimal optimal correlation clustering, the number of positive edges between $C' - N_C(u)$

and $C - (C' - N_C(u))$ is larger than $|C' - N_C(u)| \cdot |C - (C' - N_C(u))|/2$. This implies that, in average, every vertex in $C' - N_C(u)$ should have positive edges to more than half of vertices in $C - (C' - N_C(u))$. Observe that the size of $C - (C' - N_C(u))$ is at most $|C| - 0.9c'|C| = (1 - 0.9c')|C|$, in which at least $0.5|C|$ vertices belong to $N_C(u)$. Because $0.5|C|$ is at least $0.45c'|C|$ more than half of $|C - (C' - N_C(u))| \leq (1 - 0.9c')|C|$, every vertex in $C' - N_C(u)$ has $\geq 0.45c'|C|$ neighbors in $N_C(u)$ in average. Since $|C' - N_C(u)| > 0.9c'|C|$, the total number of edges between $C' - N_C(u)$ and $N_C(u)$ is at least $0.405(c')^2|C|^2 \geq 0.025|C|^2$. We then apply Janson's inequality to show that at least one of such edges will be sampled in $T$ with high probability.

Let $R$ be the set of positive edges between $C' - N_C(u)$ and $N_C(u)$ where $|R| \geq 0.025|C|^2 \geq 2.5 \cdot 10^{-6}\varepsilon^2 n^2$. By setting the family of subsets to be $R$, $X$ to be the number of positive edges $(v, w) \in R$ where both $v, w \in T$, we have $\lambda = \mathbb{E}[X] = |R| \cdot p^2$ and $\Delta \leq \frac{1}{2}|C| \cdot |R| \cdot p^3$. By applying Janson's inequality, we get

$$\Pr[\forall v, w \in T, (v, w) \notin R] \leq \exp\left(-\frac{\lambda^2}{\lambda + 2\Delta}\right)$$

When $\lambda > 2\Delta$,

$$\Pr[\forall v, w \in T, (v, w) \notin R] \leq \exp(-\lambda/2) \leq \exp(-|R| \cdot p^2/2) \leq 1/100$$

When $\lambda \leq 2\Delta$, because $|R| \geq 0.025|C|^2$,

$$\Pr[\forall v, w \in T, (v, w) \notin R] \leq \exp\left(-\frac{\lambda^2}{4\Delta}\right) \leq \exp\left(-\frac{|R|^2 p^4}{2|C||R|p^3}\right) \leq \exp\left(-0.0125p|C|\right) \leq 1/100$$

Thus, with high probability there exists $w \in N_C(u) \cap T$ and $v \in C' \cap T$ such that $u \to w \to v$ is the desired positive path. $\qquad\square$

*Proof to Theorem 1.* By Lemma E.1, Algorithm 3 always returns "YES" if $G$ is clusterable. In addition, by Lemma 3.2, Algorithm 3 returns "YES" with probability $\geq 99/100$ if $G$ is $(\varepsilon^2/10^{14})$-close-to-clusterable. What is remained is to show that Algorithm 3 will output "NO" with high probability when $G$ is far from clusterable.

To accommodate Janson's inequality, instead of working on the algorithm of sampling a fixed number of vertices, we introduce and analyze an intermediate algorithm where each vertex is included in the sample set with *independent* probability.

Let algorithm $\Pi$ follow the same step 2 and 3 as Algorithm 3. For step 1, $\Pi$ instead samples a set of vertices $S = S_1 \cup S_2 \cup S_3$, where each vertex is included in $S_1, S_2, S_3$ independently with probability $p := \min(\frac{2 \cdot 10^5}{\varepsilon n}, 1)$.

By the standard Chernoff bound and a union bound, with $\leq 1/100$ probability any of $S_1, S_2, S_3$ has a size $\geq 10^6/(3\varepsilon)$ for every $\varepsilon \in (0, 1)$. Therefore, only with $\leq 1/100$ probability Algorithm 3 samples less vertices than Algorithm $\Pi$. We instead show that Algorithm $\Pi$ has success probability $\geq 91/100$ given a graph that is $\varepsilon$-far from being clusterable.

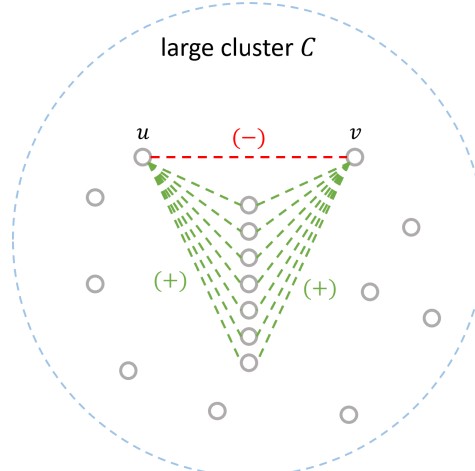 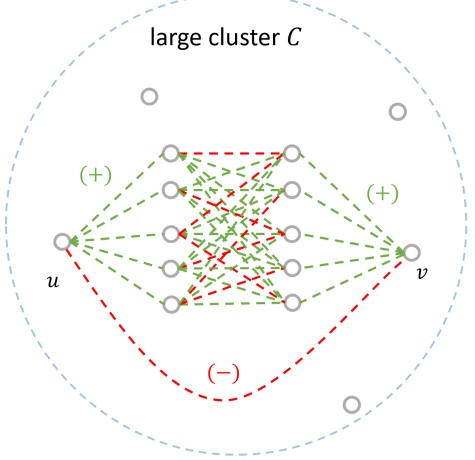

(a) $(u, v)$ is a false negative edge sampled in $S_1$, in a large cluster $C$. If $u, v$ share a large proportion of positive neighbors in $C$, with high probability $S_2$ can sample at least one of such vertices. Any of those vertices together with $u, v$ form a bad triangle.

(b) Both $u, v$ have at least half of positive neighbors in $C$ (due to the optimality of $\mathcal{C}$). When the positive neighbors of $u$ and $v$ have a small intersection, the neighbors of $u$ and $v$ roughly form a partition to $C$. A large proportion of edges between the two parts should be positive; or otherwise splitting $C$ into two parts will decrease the cost of $\mathcal{C}$. With high probability at least one of such positive edges will be sampled in $S_2$, which forms a positive path between $u$ and $v$.

Figure 3: Two subcases of Case 1. (Formalized at Lemma E.2.)

**Case 1** ($|F_N| \geq |F_P|$). $|F_N| \geq |F_P|$ implies that $|F_N| \geq \varepsilon \binom{n}{2}/2$, and the total number of (both positive and negative) edges in all the clusters

$$\sum_{i=1}^{k} \binom{|C_i|}{2} > 2|F_N| \geq \varepsilon \binom{n}{2}.$$

Because the density of $(+)$ edges inside the clusters of the optimal clustering $\mathcal{C}$ is always $> 1/2$. Or otherwise we can always further partition $\mathcal{C}$ into smaller clusters without increasing the cost.

Among these edges, at least $0.225(\varepsilon n^2 - n) > 0.224\varepsilon n^2$ of false negative edges are belonging to clusters of $\geq \frac{\varepsilon n}{20}$ vertices, since the total number of edges in clusters of small size is at most $\frac{1}{2} \cdot n \cdot \frac{\varepsilon n - 1}{20}$. We will show that the sampled vertex set $S_1$ contains at least one false negative edges $(u, v)$ with high probability; and, in addition, the induced subgraph over $S_2 \cup \{u, v\}$ contains a positive path between $u$ and $v$ with high probability. We denote by $F'_N \subseteq F_N$ the subset of false negative edges with at least one endpoint in large clusters. Since the two probabilities are dependent, we will rewrite the probability as the summation of two independent probabilities. Let $\mathcal{E}_{u,v}$ denote the event that "$u, v$ are not connected by a positive path in the induced subgraph over $S_2 \cup \{u, v\}$". We have

$$
\begin{aligned}
&\Pr[\forall u, v \in S, (u, v) \notin F'_N \vee \mathcal{E}_{u,v}] \\
\leq\ &\Pr[\forall u, v \in S_1, (u, v) \notin F'_N \vee \mathcal{E}_{u,v}] \\
\leq\ &\Pr[\forall u, v \in S_1, (u, v) \notin F'_N] + \max_{\substack{S_1 \subseteq V, u, v \in S_1: \\ (u,v) \in F'_N}} \Pr[\mathcal{E}_{u,v}].
\end{aligned}
\tag{1}
$$

The first half can be bounded using Janson's inequality. By setting the family of subsets to be $F'_N$, $X$ to be the number of pairs $(u, v) \in F'_N$ from $S_1$, we have $\lambda = \mathbb{E}[X] = |F'_N| \cdot p^2$ and $\Delta \leq \frac{1}{2} \cdot n |F'_N| \cdot p^3$.

By applying Janson's inequality, we get

$$\Pr[\forall u, v \in S_1, (u,v) \notin F'_N] = \Pr[X = 0]$$

$$\leq \exp(-\frac{\lambda^2}{\lambda + 2\Delta})$$

$$\leq \exp(-\frac{|F'_N|^2 \cdot p^4}{1.1 n |F'_N| \cdot p^3}) \tag{2}$$

$$\leq \exp(-\frac{0.224 \varepsilon p n}{1.1})$$

$$< 1/100$$

where the third inequality is by the fact that $\frac{1}{2} \cdot n |F'_N| \cdot p^3 \gg \lambda$, and the fourth inequality is due to $|F'_N| \geq 0.224 \varepsilon n^2$.

The second probability in the last line of (1) is bounded by applying Lemma E.2. When we set $C'$ to be $N_C(v)$ whose size is at least $|C|/2$, we obtain that with $\geq 99/100$ probability $u$ and $v$ are connected in the induced subgraph of $G$ over $S_2 \cup \{u, v\}$.

Therefore, for every $(u,v) \in F'_N$ and every $S_1$ that contain $u, v$, the probability that $u, v$ are not connected by a path in $G_S$ is at most $1/100$. By (1) and (2)

$$\Pr[\forall u, v \in S, (u,v) \notin F'_N \vee u, v \text{ are not connected in } G_S] \leq 2/100$$

and the probability that Algorithm 3 succeeds is at least $9/10$.

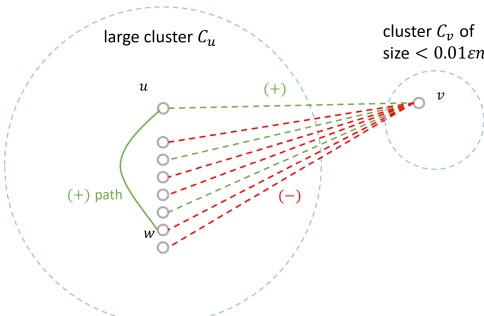
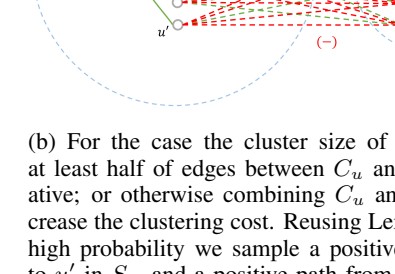

(a) $(u,v)$ is a false positive edge sampled in $S_1$. In the subcase where $|C_v| < 0.01\varepsilon n$, $u$ belongs to a large cluster $C_u$ and $v$ belongs to a small cluster $C_v$. Then $v$ will have a large number of negative edges connected to $C_u$; or moving $v$ from $C_v$ to $C_u$ will decrease the clustering cost. By a similar argument as in Figure 3 (formalized in Lemma E.2), with high probability in $S_2$ a vertex $w$ where $(w,v)$ is negative and a positive path from $u$ to $w$ are sampled. Here $(v,w)$ is the negative edge connected by a positive path $v \rightarrow u \rightarrow \cdots \rightarrow w$.

(b) For the case the cluster size of $C_v$ is $\Omega(\varepsilon n)$, at least half of edges between $C_u$ and $C_v$ are negative; or otherwise combining $C_u$ and $C_v$ will decrease the clustering cost. Reusing Lemma E.2, with high probability we sample a positive path from $u$ to $u'$ in $S_2$, and a positive path from $v$ to $v'$ in $S_3$, such that $(u', v')$ is negative. In this way, we find a negative edge $(u', v')$ connected by a positive path $u' \rightarrow \cdots \rightarrow u \rightarrow v \rightarrow \cdots \rightarrow v'$.

Figure 4: Two subcases of Case 2.

**Case 2** ($|F_N| < |F_P|$ **and** $|F_{P,L}| \geq |F_{P,S}|$)**.** By our conditions, $|F_{P,L}| \geq \varepsilon \binom{n}{2}/4$. Our proof idea to this case is similar to case 1. Through $S_1$ we will fix an edge $(u,v)$ from $F_{P,L}$, by using Janson's inequality in exactly the same way as (2). Let $C_u, C_v$ respectively denote the cluster of $u$ and $v$ in $\mathcal{C}$. Without loss of generality we assume $|C_u| \geq |C_v|$. By the definition of $F_{P,L}$, $|C_u| \geq \varepsilon n/20$. There are two subcases on whether $C_v$ is large or small.

Consider the case $|C_v| < 0.01\varepsilon n$. Let $C' := \{w \in C_u : (v,w) \in E^-\}$. Then $|C'| \geq |C_u|/10$, or otherwise moving $v$ from $C_v$ to $C_u$ will decrease the cost. By Lemma E.2, with at least $\geq 99/100$ probability the induced subgraph of $G$ over $\{u, v\} \cup S_2$ contains a path between $v$ and a vertex $w \in C_u \cap S_2$ such that $(v,w) \in E^-$.

Now we turn to the case $|C_v| \geq 0.01\varepsilon n$. Let $R_{u,v} := \{(u', v') \in E^- : u' \in C_u, v' \in C_v\}$. By the optimality of $\mathcal{C}$, $|R_{u,v}| \geq |C_u| \cdot |C_v|/2$, or otherwise combining $C_u$ and $C_v$ into a single cluster will decrease the cost. Let $Q_u \subseteq C_u$ be the set of vertices that has $\geq \frac{|C_v|}{4}$ negative neighbors in $C_v$. Then $|Q_u| \geq \frac{|C_u|}{3}$ since otherwise $|R_{u,v}| < |C_u| \cdot |C_v|/2$. By applying Lemma E.2, the induced subgraph over $S_2 \cup \{u\}$ contains a positive path from $u$ to $Q_u$ with $\geq 99/100$ probability. Fix the vertex sampled in $Q_u$ as $u'$, we denote $Q_v \subseteq C_v$ as the subset of $C_v$ whose vertices have negative edges to $u'$. By our definition to $Q_u$, $|Q_v| \geq \frac{|C_v|}{4}$. Again by applying Lemma E.2, the induced subgraph over $S_3 \cup \{v\}$ contains a positive path from $v$ to $Q_v$ with $\geq 99/100$ probability. We denote by $v'$ the vertex sampled in $Q_v$.

Therefore, with probability $\geq 97/100$ there exists a negative edge $(u', v')$ where $u', v' \in S$ such that $u', v'$ are connected by a positive path

$$u' \rightarrow \cdots \rightarrow u \rightarrow v \rightarrow \cdots \rightarrow v'$$

in the induced subgraph over $S$. Therefore, Algorithm 3 is correct with high probability.

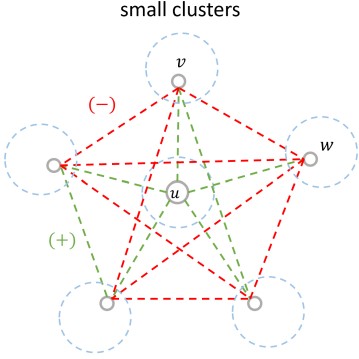

Figure 5: In Case 3, with high probability, the sample set $S_1$ contains a vertex $u$ that belongs to a small cluster, and is connected to $\Omega(\varepsilon n)$ vertices of small clusters by positive edges. The induced subgraph over all such positive neighbors of $u$ must contain a bounded proportition ($\leq 0.94$) of positive edges, or otherwise forming these vertices into a single cluster will decrease the total cost. Therefore, with high probability a negative edge $(v, w)$ will be sampled in $S_2$, which forms a bad triangle $(u, v, w)$.

**Case 3** ($|F_N| < |F_P|$ **and** $|F_{P,L}| < |F_{P,S}|$)**.** By our conditions, $|F_{P,S}| > \varepsilon \binom{n}{2}/4$. There are at least $0.01\varepsilon n$ vertices that are incident to $\geq 0.23\varepsilon n$ edges from $|F_{P,S}|$. By Chernoff bound, at least one of such vertices $u$ is sampled in $S_1$ with $\geq 99/100$ probability. Define

$$N_{u,S} := \{v : (u, v) \in E^+, u \not\sim v, \text{ and } v \text{ belongs to a cluster of size } (< \varepsilon n/20)\}.$$

Then $|N_{u,S}| \geq 0.23\varepsilon n$. We will show that the density of positive edges in the induced subgraph of $G$ over $N_{u,S}$ is small, or otherwise $N_{u,S}$ can form a cluster with smaller cost.

Since all the vertices in $N_{u,S} \cup \{u\}$ belong to clusters of size at most $\varepsilon n/20$, splitting all these vertices from their clusters will at most increase $(|N_{u,S}| + 1) \cdot \varepsilon n/20$ cost. Thus, the total number of negative edges inside $N_{u,S} \cup \{u\}$ is at least

$$\frac{1}{2}\binom{|N_{u,S}| + 1}{2} - (|N_{u,S}| + 1) \cdot \varepsilon n/20 = (|N_{u,S}| + 1) \cdot \frac{5|N_{u,S}| - \varepsilon n}{20} > \frac{|N_{u,S}| \cdot \varepsilon n}{31}.$$

Otherwise, making $N_{u,S} \cup \{u\}$ a cluster will decrease the cost of $\mathcal{C}$ by calculating its relative cost.

We will again use Janson's inequality to show that at least one of such non-edges $(v, w)$ will be sampled in $S_2$ with high-probability. Since $(u, v), (u, w)^+ \in E$ but $(v, w) \in E^-$, a bad triangle is observed.

Let $R$ be the set of these non-edges, where $|R| > |N_{u,S}| \cdot \varepsilon n/31$. Let $X$ be the random variable denoting the number of non-edges in $R$ that are included in $G_{S_2}$. Let $\lambda = \mathbb{E}[X] = |R| \cdot p^2$. Let

$\Delta = \frac{1}{2}|N_{u,S}| \cdot |R| \cdot p^3$. By Janson's inequality, we have

$$\Pr[X = 0] \leq \exp\left(-\frac{|R| \cdot p}{1.1 \cdot |N_{u,S}|}\right) \leq 1/100.$$

Therefore, with $\geq 98/100$ probability one can observe vertices $v, w \in S_2$ and $u \in S_1$ such that $(v, w) \notin E$ but $(u, v), (u, w) \in E$ in this case.

To summarize, in all of the above three cases, one can always observe a bad triangle with probability $\geq 9/10$ probability when the graph is $\varepsilon$-far from being clusterable. Algorithm 3 outputs correctly with probability $\geq 9/10$.

$\square$

## F  OPTIMAL STRUCTURAL BALANCE TESTING FOR COMPLETE GRAPHS

We now discuss our results for structural balance, i.e., the case of $k = 2$ for correlation clustering with a fixed number of clusters. Recall that the main theorem statement is as follows.

**Theorem 3.** *Fix $\varepsilon \in (0, 1)$. There exists a randomized algorithm that given a labeled complete graph $G = (V, E^+ \cup E^-)$ and a parameter $\varepsilon$ answers the following*

- *If $G$ is balanced, the algorithm always answers "YES";*

- *If $G$ is at least $\varepsilon$-far from being balanced, the algorithm answers "NO" with probability $\geq 0.9$.*

*The algorithm queries at most $O(1/\varepsilon)$ edges of $G$ and runs in $O(1/\varepsilon)$ time.*

The algorithm for Theorem 3 uses a different approach compared to Theorem 2: we sample $O(1/\varepsilon)$ triangles, take the graph $G'$ induced by the edges (the graph might *not* be complete), and check whether $G'$ has any unbalanced triangle. The formal algorithm could be described as follows.

---

**Algorithm 4.  An algorithm for structural balance property testing**
**Input:** A labeled complete graph $G = (V, E^+ \cup E^-)$; a parameter $\varepsilon$.

1. Sample $300/\varepsilon$ triangles $(u, v, w) \in V^3$ uniformly at random (with replacement).
2. Check if any of the sampled triangles is unbalanced.

---

We first observe that Algorithm 4 uses $O(1/\varepsilon)$ queries and time, and the algorithm always returns "*balanced*" if $G$ is indeed balanced.

**Lemma F.1.** *Algorithm 4 makes $O(1/\varepsilon)$ queries to $G$ with $O(1/\varepsilon)$ computation time.*

*Proof.* The algorithm only samples $O(1/\varepsilon)$ edges and triangles, where we use $O(1)$ time for each triangle to check whether it is balanced or not. $\square$

**Lemma F.2.** *If $G = (V, E^+ \cup E^-)$ is balanced, then Algorithm 4 always returns "balanced".*

*Proof.* By a simple observation, any subgraph of a (strongly) balanced graph does not contain any unbalanced triangle. Therefore, the algorithm will *not* detect any unbalanced triangle and will always return "*balanced*". $\square$

We now proceed with the proof of the soundness of the algorithm. At a high level, we aim to demonstrate that if the number of disagreement edges is high, then the number of unbalanced triangles has a similar lower bound. Proving the statement, however, is not entirely straightforward since the number of unbalanced triangles is not necessarily monotone w.r.t. the number of flipped edges – it depends on the structure of the graph. Consider, for instance, a graph of $n$ vertices with exactly two false edges. If the two edges are not incident to each other, the total number of unbalanced triangles is $2(n-2)$, $n-2$ unbalanced triangles induced by each false edge and each other vertex. But if the

two edges are incident to each other, the triangle including both edges will be balanced, and the total number of unbalanced triangles decrease to $2(n-3)$. We give a clean proof to the desired statement using three different random sampling processes, avoiding discussions to the structure of the graph.

**Lemma F.3.** *If $G = (V, E^+ \cup E^-)$ is at least $\varepsilon$-far from being balanced, then Algorithm 4 returns* "not balanced" *with probability at least* $99/100$.

*Proof.* Let $\mathcal{X}_{\text{unbalanced}}$ be the set of unbalanced triangles in $G$, and let $E_{\text{unbalanced}}$ be the set of false edges induced by $(L^*, R^*)$, which is the optimal partition that minimizes the frustration index of the graph. For each unbalanced triangle $\Delta \in \mathcal{X}_{\text{unbalanced}}$. We define the following sampling process for triangles.

---

**Process 1:** a random sampling process for triangles.

- Sample a triangle uniformly at random from $G$.

---

Let $X_\Delta$ be the indicator random variable for the unbalanced triangle $\Delta \in \mathcal{X}_{\text{unbalanced}}$ to be sampled by *Process 1*.

We now consider another sampling process in which we uniformly sample an edge.

---

**Process 2:** a random sampling process for edges.

- Sample an edge uniformly at random from $G$.

---

For each false edge $e \in E_{\text{unbalanced}}$, we let $Y_e$ be the indicator random variable for the edge to be sampled. Our technical claim is as follows.

**Claim F.4.** *On each time of sampling with* Process 1 *and* Process 2*, we have that*

$$\Pr\left(\Delta \in \mathcal{X}_{\text{unbalanced}} \text{ is sampled by Process 1}\right) \geq \frac{1}{3} \Pr\left(e \in E_{\text{unbalanced}} \text{ is sampled by Process 2}\right).$$

*Proof.* We consider the following sampling process:

---

**Process 3:** a "bridge" sampling process.

- Sample a vertex $v \in V$ uniformly at random;

- Sample an edge $e' \not\ni v$ uniformly at random.

---

We let $(L^v, R^v)$ be the partition obtained by the following rules: we arrange all the $(+)$ neighbors of $v$ in $L^v$. The set of the rest of the vertices, namely $V \setminus L^v$, is therefore defined as $R^v$. Let $E_{\text{unbalanced}}(v)$ be the set of false edges induced by $(L^v, R^v)$. Since $(L^*, R^*)$ is the optimal partition that minimizes the frustration index, for any $v \in V$, we have that

$$|E_{\text{unbalanced}}(v)| \geq |E_{\text{unbalanced}}|.$$

Therefore, conditioning on the sampling of any fixed $v$, we have that

$$\Pr\left(e \in E_{\text{unbalanced}} \text{ is sampled by Process 2}\right) \leq \Pr\left(e \in E_{\text{unbalanced}}(v) \text{ is sampled by Process 3}\right).$$

On the other hand, for any fixed $v$, let $\mathcal{X}_{\text{unbalanced}}(v)$ be the set of unbalanced triangles with one endpoint as $v$ and one edge in $E_{\text{unbalanced}}(v)$. Note that each unbalanced triangle will at most be counted 3 times, which happens only when all the three edges of the triangle are false edges. As such, we have that

$$3 |\mathcal{X}_{\text{unbalanced}}| \geq \sum_{v \in V} |\mathcal{X}_{\text{unbalanced}}(v)|.$$

Therefore, we could lower bound the probability of sampling an unbalanced triangle as

$$\Pr\left(\Delta \in \mathcal{X}_{\text{unbalanced}} \text{ is sampled by Process 1}\right)$$

$$\geq \frac{1}{3} \sum_{v \in V} \Pr\left(v \text{ is sampled and } \Delta \in \mathcal{X}_{\text{unbalanced}}(v) \text{ is sampled by Process 3}\right)$$

$$= \frac{1}{3} \sum_{v \in V} \Pr\left(v \text{ is sampled by Process 3}\right) \cdot \Pr\left(e \in E_{\text{unbalanced}}(v) \text{ is sampled by Process 3}\right).$$

Observe that each vertex has $1/n$ probability to be sampled in Process 3. Therefore, we have

$$\Pr\left(\Delta \in \mathcal{X}_{\text{unbalanced}} \text{ is sampled by Process 1}\right)$$

$$\geq \frac{1}{3} \sum_{v \in V} \frac{1}{n} \cdot \Pr\left(e \in E_{\text{unbalanced}}(v) \text{ is sampled by Process 3}\right)$$

$$\geq \frac{1}{3} \sum_{v \in V} \frac{1}{n} \cdot \Pr\left(e \in E_{\text{unbalanced}} \text{ is sampled by Process 2}\right)$$

$$= \frac{1}{3} \Pr\left(e \in E_{\text{unbalanced}} \text{ is sampled by Process 2}\right),$$

which is as desired by the statement. Lemma F.4 □

Since the graph is at least $\varepsilon$-far from being balanced, we have that

$$\Pr(e \in E_{\text{unbalanced}} \text{ is sampled by Process 2}) \geq \varepsilon.$$

Therefore, by Lemma F.4, for each time of sampling in Algorithm 4, we have that

$$\Pr\left(X_\Delta = 1 \text{ for some } \Delta \in \mathcal{X}_{\text{unbalanced}}\right) \geq \varepsilon/3.$$

Since we sample triangles without replacement, the sampling at each time is independent. Therefore, the probability for us to not sample any unbalanced triangle with $100/\varepsilon$ samples is at most $(1 - \varepsilon/3)^{300/\varepsilon} \leq 1/100$, as desired by the lemma statement. Lemma F.3 □

Combining Lemma F.1, Lemma F.2, and Lemma F.3 gives the full proof of Theorem 3.

## G    EXTENSION TO STRUCTURAL BALANCE TOLERANT TESTING

We now discuss generalizing our structural balance testing algorithm to allow graphs that are *nearly balanced* acceptable by the tester. This falls into the regime of *tolerant testing* Parnas et al. (2006); Ron (2009); Blais et al. (2019), in which we want instances that *nearly* satisfied the desired property to also pass the test. For structural balance, a testing algorithm as such has strong practical motivations: real-world graphs are often *not perfectly balanced* yet *close to being balanced*. Therefore, a tolerant testing algorithm could have a much broader impact on testing read-world graphs.

Recall that main theorem for the tolerant testing algorithm is as follows.

**Theorem 4.** *Fix $\varepsilon \in (0, 1)$ such that $\delta \leq \varepsilon/900$. There exists a randomized algorithm that given a labeled complete graph $G = (V, E^+ \cup E^-)$ and parameters $\varepsilon, \delta$ answers the following*

- *If $G$ is at most $\delta$-close from being balanced, the algorithm answers "YES" with probability $\geq 0.99$;*

- *If $G$ is at least $\varepsilon$-far from being balanced, the algorithm answers "NO" with probability $\geq 0.99$.*

*The algorithm queries at most $O(1/\varepsilon)$ edges of $G$ and runs in $O(1/\varepsilon)$ time.*

The algorithm for Theorem 4 is similar to Algorithm 4, albeit we use a *threshold* to determine whether the graph is balanced. The algorithm could be described as follows.

---

**Algorithm 5. An algorithm for structural balance tolerant testing**
**Input:** A labeled complete graph $G = (V, E^+ \cup E^-)$, parameters $\varepsilon, \delta$ such that $\delta \leq \varepsilon/900$.

1. Sample $300/\varepsilon$ triangles $(u, v, w) \in V^3$ uniformly at random (with replacement).
2. If at most 10 out of $300/\varepsilon$ sampled triangles are unbalanced, return "balanced". Otherwise, return "not balanced".

---

The efficiency of the algorithm follows directly from the argument as in Lemma F.1, and we write the corresponding lemma without proof.

**Lemma G.1.** *Algorithm 5 makes $O(1/\varepsilon)$ queries to $G$ and converges in $O(1/\varepsilon)$ time.*

We first analyze the *soundness* of the algorithm, for which we could use the conclusion in Lemma F.4. The main lemma is as follows.

**Lemma G.2.** *If $G = (V, E^+ \cup E^-)$ is at least $\varepsilon$-far from being balanced, then Algorithm 5 returns "not balanced" with probability at least $199/200$.*

*Proof.* Let $X_\Delta$ be the indicator random variable for an unbalanced triangle $\Delta$ to be sampled for one sampling step in Algorithm 5, and let $X = \sum X_\Delta$ be the total number of unbalanced triangles sampled by Algorithm 5. By Lemma F.4, we have that

$$\mathbb{E}[X] = \frac{300}{\varepsilon} \cdot \Pr(X_\Delta = 1 \text{ for some } \Delta \in \mathcal{X}_{\text{unbalanced}}) \geq 100.$$

Since $X$ is a summation of independent indicator random variables, we could apply Chernoff bound, and show that

$$\Pr(X \leq 10) \leq \Pr(X \leq (1 - 0.9) \cdot \mathbb{E}[X])$$
$$\leq \exp\left(-\frac{0.9^2 \cdot 100}{2}\right) \leq 1/200,$$

as desired. $\qquad\square$

We now proceed to show the completeness of the algorithm, i.e., $\delta n^2$-close instances are also to pass the test and result in a "*balanced*" outcome. The proof of the lemma will use a "reversed" probability calculation as in Lemma F.4.

**Lemma G.3.** *If $G = (V, E^+ \cup E^-)$ is at most $\delta$-far from being balanced for some $\delta \leq \varepsilon/900$, then Algorithm 4 returns "balanced" with probability at least $199/200$.*

*Proof.* Similar to the proof of Lemma F.3, we let $\mathcal{X}_{\text{unbalanced}}$ be the set of unbalanced triangles, and $E_{\text{unbalanced}}$ be the set of disagreement edges induced by the optimal partition $(L^*, R^*)$. We now define the following processes.

---

**Process 4:** a random sampling process for edges.

- Sample a triangle $\Delta$ uniformly at random from $G$, then sample an edge from $\Delta$.

---

Also, we will use the random Process 1 which samples a triangle uniformly at random from the graph (see the proof of Lemma F.3 for the full description). We have the following technical claim.

**Claim G.4.** *On each time of sampling with* Process 1 *and* Process 4*, we have that*

$$\Pr(\Delta \in \mathcal{X}_{\text{unbalanced}} \text{ is sampled by Process 1}) \leq 3 \cdot \Pr(e \in E_{\text{unbalanced}} \text{ is sampled by Process 4}).$$

*Proof.* The claim follows from the fact that for each $\Delta \in \mathcal{X}_{\text{unbalanced}}$, there must be at least one edge $e \in E_{\text{unbalanced}}$ *by definition*. Therefore, we have that

$\Pr(e \in E_{\text{unbalanced}} \text{ is sampled by Process 4})$

$\geq \Pr(\text{sampling } e \in E_{\text{unbalanced}} \text{ from } \Delta \in \mathcal{X}_{\text{unbalanced}}) \cdot \Pr(\Delta \in \mathcal{X}_{\text{unbalanced}} \text{ is sampled by Process 1})$

$\geq \frac{1}{3} \cdot \Pr(\Delta \in \mathcal{X}_{\text{unbalanced}} \text{ is sampled by Process 1}),$

which leads to the desired statement. Lemma G.4 $\square$

For a graph that is at most $\delta$-far from being balanced, which means it is at most $(\varepsilon/900)$-far from being balanced, we have that

$$\Pr(e \in E_{\text{unbalanced}} \text{ is sampled by Process 2}) \leq \frac{\varepsilon}{900}.$$

Let $X = \sum X_\Delta$ be the total number of unbalanced triangles sampled by Algorithm 5. By Lemma G.4, we have that

$$\mathbb{E}[X] = \frac{300}{\varepsilon} \cdot \Pr(X_\Delta = 1 \text{ for some } \Delta \in \mathcal{X}_{\text{unbalanced}})$$

$$\leq \frac{900}{\varepsilon} \cdot \Pr(e \in E_{\text{unbalanced}} \text{ is sampled by Process 2}) \leq 1.$$

If $X < 1$, then Algorithm 5 deterministically returns "*balanced*". As such, we assume w.log. that $X \geq 1$. Since $X$ is a summation of independent indicator random variables, we could apply the Chernoff bound, and we get

$$\Pr(X \geq 10) \leq \Pr(X \leq (1+9) \cdot \mathbb{E}[X])$$

$$\leq \exp\left(-\frac{81 \cdot 1}{11}\right) \leq 1/200,$$

as desired. $\square$

Combining Lemma G.1, Lemma G.2 and Lemma G.3 with a union bound gives the desired statement of Theorem 4.

# H A LOWER BOUND FOR TESTING CLUSTERABILITY AND STRUCTURAL BALANCE

We give a lower bound for testing structural balance in complete graphs in this section. Our lower bound shows that any algorithm that separates a graph from being balanced vs. $\varepsilon$-far from being balanced requires at least $\Omega(1/\varepsilon)$ queries to the graph. This implies our algorithms in Theorem 3 and Theorem 4 are asymptotically optimal.

Recall that our statement for the lower bound is as follows.

**Theorem 5.** *Any (possibly randomized) algorithm that given a complete labeled graph $G = (V, E^+ \cup E^-)$, with probability at least $2/3$ answers correctly whether $G$ is balanced or at least $\varepsilon$-far from being balanced requires at least $\Omega(1/\varepsilon)$ edge queries to the graph.*

*Furthmore, the lower bound extends to testing clusterability (for both general $k$ and fixed $k$).*

*Proof.* We use the following result from a recent paper to prove our lower bound.

**Proposition H.1** (Fischer (2024), rephrased; cf. Bshouty & Goldreich (2025)). *Let $\Sigma$ be an arbitrary alphabet for an length-$m$ input, and let $\Sigma^m$ be the set of all possible inputs. Let $\mathcal{P} \subseteq \Sigma^m$ be the set of inputs that satisfy a property. Suppose there exists an instance $U \notin \mathcal{P}$ such that at least $\alpha \cdot m$ elements need to be modified to satisfy the property prescribed by $\mathcal{P}$. Then, any algorithm that with probability at least $2/3$ correctly distinguishes whether an input $S \in \Sigma^m$ is in $\mathcal{P}$ or needs to modify at least $\beta \cdot m$ bits to satisfy property of $\mathcal{P}$ requires $\Omega(\alpha/\beta)$ queries to $S$.*

We apply Lemma H.1 with $\Sigma = \{(+), (-)\}$ and $m = \binom{n}{2}$. The instances with structural balance are $\mathcal{P}$. Here, we only need to find an instance $U \notin \mathcal{P}$ at least $\alpha$-far from being balanced for some $\alpha = \Omega(1)$. We consider a graph with all $(-)$ edges as such an instance: the graph has $\binom{n}{3}$ bad triangles, and each flip of the label could reduce the number of bad triangles by at most $n - 1$. As such, the graph is at least $\alpha$-far from being balanced for some $\alpha = \Omega(1)$. Applying Lemma H.1 leads to the desired $\Omega(1/\varepsilon)$ query lower bound. $\square$

Note that since each query takes $O(1)$ time, our algorithms are also asymptotically optimal in terms of the time complexity.

