# OpenReview forum: "Efficient Testing for Correlation Clustering: Improved Algorithms and Optimal Bounds"
_ICLR.cc/2026/Conference — ICLR 2026 Poster_

### Official Review · Reviewer_Yvv8 · 2025-10-31

**Soundness:** 3
**Presentation:** 3
**Contribution:** 3
**Rating:** 8
**Confidence:** 3

**Summary:**

This paper studies the problem of testing the cost of correlation clustering, i.e., to test whether the input graph $G=(V,E^+\cup E^-)$ is clusterable (with $0$ cost) or is $\varepsilon$-far away from being clusterable. Here, the cost of correlation clustering is defined to be the number of $(+)$ edges crossing different clusters and the number of $(-)$ edges in the same cluster. $\varepsilon$-far away from being clusterable means that one have to flip the labels of at least $\varepsilon\cdot \binom{n}{2}$ edges to make the graph clusterable, where $n=|V|$.

The authors proposed new testers that achieve improved query complexity compared to prior work. Specifically,
* for correlation clustering, they gave a new tester with $O(\frac{1}{\varepsilon^2})$ query complexity while  previous best result is $\widetilde{O}(\frac{1}{\varepsilon^7})$,
* for correlation clustering with fixed $k\ge 2$, they give the the first nontrivial tester with $O(\frac{k^4\ln ^4 k}{\varepsilon^4})$ query complexity,
* for correlation clustering with $k=2$ (i.e., structural balance), they gave a new tester with $O(\frac{1}{\varepsilon})$ query complexity while  previous best result is $\widetilde{O}(\frac{1}{\varepsilon^2})$ (all the time complexities are proportional to their query complexity).
* Moreover, to complement their upper bounds, they gave an $\Omega(\frac{1}{\varepsilon})$ query complexity lower bound for structural balance problem. The proof of lower bound relies on a direct application of an existing general lower bound result (Fischer, 2024; cf. Bshouty & Goldreich, 2025) to the problem of testing structural balance.

Technically, they employ sampling-based techniques combined with Janson’s inequality to analyze the concentration of local inconsistencies, a methodological novelty compared to classical approaches using the graph removal lemma. Experiments on synthetic and real-world datasets (from SNAP) confirm the theoretical query and runtime improvements.

**Strengths:**

* This paper studies a problem that is well-motivated.

* This paper improves the query complexity of testing the cost of correlation clustering from $\widetilde{O}(\frac{1}{\varepsilon^7})$ to $O(\frac{1}{\varepsilon^2})$, which is a substantial and clear advancement over the state of the art.

* For structural balance, the authors prove the first tight $O(\frac{1}{\varepsilon})$ bounds for testing structural balance.

* The use of Janson’s inequality for analyzing property testers in labeled graphs is elegant and potentially applicable to other testing problems.

* Experiments on both synthetic and real-world graphs show practical improvements in query complexity and runtime, enhancing the paper’s credibility.

* This paper is well written.

**Weaknesses:**

* While the theoretical improvement is substantial, the algorithms themselves are relatively straightforward extensions of uniform sampling ideas, and much of the novelty lies in tighter analysis rather than fundamentally new algorithmic constructs.

* The results for general correlation clustering and correlation clustering with fixed $k$ are not tight ($O(\frac{1}{\varepsilon^2})$ vs. $O(\frac{1}{\varepsilon})$ and $O(\frac{1}{\varepsilon^4})$ vs. $O(\frac{1}{\varepsilon})$, respectively). The authors acknowledge this but do not deeply explore whether the bound can be improved.

* Nevertheless, I am generally supportive of this work.

**Questions:**

* lines 357-358: why there is a $2$ in $\sum_{1\le i<j\le s}{2\cdot \frac{1}{n^2}}$?

* about the proof of Lemma 3.1: It samples $O(\frac{k\ln k}{\varepsilon})$ vertices and for each iterated vertex $u$, it will perform at most $k$ queries. Why is the total number of queries $O(\frac{k\ln k}{\varepsilon})$ rather than $O(\frac{k^2\ln k}{\varepsilon})$?

**Typos:**

* lines 362-363: missing a period after the inequality

* lines 756-757: we may assume that $t>k$, since a graph ... $\rightarrow$ we may assume that $t>k$, since **otherwise** a graph ...?

---

> ### Author Response · Authors · 2025-11-24
>
> We are grateful to the reviewer for the valuable feedback. We would like to give responses to your comments and questions.
>
> ### **Response to “The algorithms are relatively straightforward extensions, not fundamentally new”**
>
> We agree with the reviewer that the algorithm is relatively straightforward. However, we would like to refer to an observation by prior works such as Goldreich and Trevisan [FOCS’01]: **Most graph property testing problems admit ‘canonical testers’**, i.e., an algorithm that samples a certain number of vertices from $V$, tests the property on the sampled subset, and returns the answer accordingly. Therefore the key technical challenge falls into a tighter analysis. Our algorithms for testing correlation clustering indeed employs this canonical tester with fewer samples, and our analysis tackles the technical barrier by new insights and the use of Janson’s inequality.
>
> Furthermore, we believe simplicity is a strength rather than a weakness. For modern large-scale algorithms in ML, simplicity ensures that the algorithm can be easily implemented by practitioners, and the provable guarantees ensure the quality of the solution. Our algorithms satisfy both properties.
>
> **Revision action:** In the revised version of the paper, we have added a discussion on the observation of the canonical tester, and the main reason our results are not known before. (line 138)
>
> ### **Response to “the results are not tight”**
>
> Thank you for pointing out the gap. Pursuing tight bounds in general correlation clustering is an interesting future direction. The lower bound technique we used for the structural balance testing (k=2) does not extend to the other settings. We are not aware of existing techniques that give a reduction on the correlation clustering testing problems.
>
> We kindly remark that we had included discussions about potential directions to make the bounds tighter in Section 5. We **expanded the discussion** in the updated version of the paper.
>
> ### **Response to “questions and typos”**
>
> Thank you for the valuable questions and careful corrections. We updated the paper according to all the questions and typos you raised. Regarding your question 1 at line 369, the factor of 2 accounts for two different orders a pair of vertices appears in the sample set ($u$ appears first, or $v$ appears first).

---

> > ### Comment · Reviewer_Yvv8 · 2025-11-27
> >
> > Thank the authors for your responses. My questions regarding the proof have now been addressed. Overall, I will maintain my rating and take a supportive stance towards this paper.

---

> > > ### Author Response · Authors · 2025-11-27
> > >
> > > We appreciate your review and support. Thank you and always feel free with any discussion.

---

### Official Review · Reviewer_CjKd · 2025-10-31

**Soundness:** 3
**Presentation:** 2
**Contribution:** 3
**Rating:** 8
**Confidence:** 3

**Summary:**

This paper studies the correlation clustering (CC) problem from the perspective of property test. Theoretically, it yields three results for clusterability, $k$-clusterability ($k$ is a constant), and $2$-clusterability (also called strong balanceness), respectively. The query complexity $O(1/\varepsilon^2)$ for clusterability gets a great improvement from the SOTA results using $\tilde{O}(1/\varepsilon^7)$. The query complexity $O(1/\varepsilon^4)$ for $k$-clusterability is new, and that for $2$-clusterability $O(1/\varepsilon)$ has reached the tight lower bound up to a constant factor. The testing algorithms are evaluated by experiments on simulated and real-world datasets.

**Strengths:**

1. The theoretical results are solid.

2. It provides more insights to the CC testing problem (e.g., the hit of bad triangles and the use of Janson's inequality).

**Weaknesses:**

1. I am skeptical about the practical merits of the CC problem. Requiring a complete graph as input is an overly stringent condition that is rarely satisfied in real-world scenarios. While the authors have presented experimental results on real-world datasets, it remains unclear how they converted the six real-world graphs into complete ones, nor how they assigned +/- signals to missing edges. So I am uncertain about the significance of these experiments. I acknowledge that this paper has theoretical significance. If I were reviewing it for a TCS venue like SODA or STOC, I would not raise this concern. However, for an AI venue, clarifying its practical relevance is essential.

   A more interesting variant of the CC problem appears to take a general graph as input and count the misclassified edges. However, the methodology to address this variant would be entirely different.

2. There are some unclear points and minor issues. Please refer to the Questions.

**Questions:**

1. The definitions of some key concepts are missing. I didn't find the formal definition of "structural balance". The authors have defined the weak and strong versions of structural balance in the Introduction section. It seems that structural balance refers specifically to $2$-clusterable, isn't it? The definition of "bad triangles" is also missing.

2. The figures in Section 4.2 is confusing. What's the difference between the blue and orange lines? Or say, what is the difference between structural balance and correlation clustering? Do they have the same number of clusters? Does a YES answer mean "clusterable"? Why its rate is nearly zero small (especially for structural balance) when the distance parameter $\varepsilon$ is small?

3. It is better to introduce Janson's inequality in a separate section in appendices for the readers who are not familiar with it.

---

> ### Author Response · Authors · 2025-11-24
>
> ### **Response to “Requiring a complete graph as input is an overly stringent condition that is rarely satisfied in real-world scenarios. It remains unclear how they converted the six real-world graphs into complete ones, nor how they assigned +/- signals to missing edges.”**
>
> Thank you for raising the concern.  For an input graph $G=(V,E)$ not necessarily complete, we can run the algorithm by treating the edges as $(+)$ edges and the non-edges as $(-)$ ones. This is exactly what we did for the experiments. Although this strategy does not fit all application scenarios, for applications like social networks, it makes perfect sense (for instance, in the college dataset, $u,v$ has a $(+)$ edge if they have message exchanges and an $(-)$ edge otherwise).
>
> **Revision action:** In the updated version, we have clarified the experiment setting and the way we acquire $(+)$ and $(-)$ edges
>
> ### **Response to “No formal definition of structural balance. The definition of bad triangles is also missing.”**
>
> We apologize for the confusion. The reviewer is correct that structural balance is mathematically equivalent to correlation clustering with 2 clusters. Furthermore, a “bad triangle” is defined as a triangle with two $(+)$ edges and one $(-)$ edge.
>
> **Revision action:** In the updated version, we have added the definitions for structural balance and bad triangles (Section 2). Thanks for the catch!
>
>
> ### **Response to “​​The figures in Section 4.2 are confusing” and “Introduce Janson's inequality”.**
>
> We thank the reviewer for raising the clarity concerns.
>
> In the experiment, the blue curves represent the testing results for structure balance (i.e., whether the graph is clusterable with 2 clusters) and the orange curves represent the testing results for clusterability (i.e., whether the graph is clusterable with any number of clusters). A graph can be clusterable without being structurally balanced, e.g., it contains a structure of 3 clusters. Therefore, as we can observe from the plots, it is always harder for the blue curve to report “YES” answers. Here, a “YES” answer means that the algorithm believes the graph is balanced (resp. clusterable), and a “NO” answer means the algorithm believes the graph is far from being balanced (resp. clusterable).
>
> The number of clusters for structural balance tests is always 2, and the number of clusters for the clusterability test does not need to be specified. Since these graphs are real-world graphs, their structures are always fixed, and they are not perfectly clusterable or balanced. Therefore, when $\varepsilon$ value is small, the algorithm almost always detects enough number of bad triangles to determine that the input is not balanced (or clusterable).
>
> Also, thanks for pointing out the missing definition of Jason’s inequality! We have added it in the updated version of the paper (in Appendix D).

---

### Official Review · Reviewer_ENS9 · 2025-11-01

**Soundness:** 3
**Presentation:** 3
**Contribution:** 2
**Rating:** 4
**Confidence:** 3

**Summary:**

The paper presents property-testing algorithms for correlation clustering and balance property in graphs. Compared to previous work, the paper presents improved bounds, and in addition it presents a new sampling algorithm for testing k-clusterability, which as far as I understand, is a newly introduced problem. The paper gives a good overview of previous results, states its contributions clearly, provides a sketch techniques while full proofs are in the appendix, and presents an empirical evaluation of the methods.

**Strengths:**

S1. Solid problem formulation, improving and extending earlier work.
S2. Theoretically rigorous paper, giving a good intuition of the techniques and presenting full proofs in the appendix.
S3. Well written paper, presenting a clear motivation, highlighting the contributions and main results, and presenting the methods with clarity.
S4. Theoretical results are accompanied with an empirical evaluation.

**Weaknesses:**

W1. While O(1/epsilon^2) is a significant improvement over \tilde{O}(1/epsilon^7), I found that the two results are not directly comparable because the earlier work of AA2023 tests of cost smaller than epsilon/10, while the current work tests for cost=0. So, I think that the current paper solves a much easier problem. Furthermore, this limitation is significant in practice, as the method essentially does not tolerate any noise, which is highly unrealistic for real-world applications. I think that the author(s) should have been more clear about this important distinction, during the discussion and when presenting Table 1. This weakness is my main motivation for not suggesting the paper for acceptance.
W2. While researching for sublinear-time methods, I think that in practice a linear algorithm is often good enough. In this sense, I think that the paper would be a better fit for a venue in theoretical computer science.

**Questions:**

I would be very interested in seeing the response of the author(s) in point W1, above.

---

> ### Author Response · Authors · 2025-11-24
>
> We thank the reviewer for the insightful review. We would like to address the raised concerns, and please check our response as follows.
>
> ### **Response to “Two results are not directly comparable because the earlier work of AA2023 tests of cost smaller than epsilon/10, while the current work tests for cost=0.”**
>
> Thank you for pointing out the concern. We would like to clarify that our algorithms are actually tolerant testers, i.e., $\text{OPT}$ being $\alpha$-close to $0$ vs. $\beta$-far from $0$.
>
> - For structural balance, in Theorem 4 (details in Appendix F), we proved that our algorithm is able to distinguish graphs that are at most $\delta$-close to being balanced vs. at least $\varepsilon$-far from being balanced for $\delta=C\cdot \varepsilon$ for some constant $C<1$.
>
> - For general correlation clustering, our algorithm can distinguish graphs that are $\varepsilon$-far from being clusterable vs. $O(\varepsilon^2)$-close to being clusterable, although we did not explicitly state it due to the context organization. Using the same proof idea as we did for the proof of Theorem 2 (between lines 355 and 365), we can show that if the graph is at most $\varepsilon^2$-close from being balanced, then no flipped edge will be sampled, which means the algorithm will return ‘YES’ in such a case.
>
> **We believe the reviewer’s main concern is on the general correlation clustering, prior work is a tolerant tester while ours is not. By response above, we clarify that our tester is also tolerant.** Further, note that prior work uses a black-box reduction, while our analysis uses new ideas on sampling positive paths and Janson’s inequality, which is non-trivial. We hope our technical contribution sheds light on a new analysis approach to broader testing problems.
>
> Finally, regarding the concern about practical performance, our experiments have shown that our algorithm indeed demonstrates some properties for tolerant testing. In the experiments with real-world graphs, the graphs are not balanced; however, when we have larger $\varepsilon$, the algorithm still output ‘YES’ (the graph is clusterable or balanced), and there is a clear phase transition between ‘YES’ and ‘NO’ answers with different $\varepsilon$ values (we did add some light algorithm engineering by setting a heuristic threshold).
>
> **Revision action:** In the updated version of the paper, we have updated our statement and the proof for the testing algorithm for clusterability to include the tolerance testing property.
>
> ### **Response to “In practice a linear algorithm is often good enough”**
>
> We thank the reviewer for the remark. We agree with the reviewer that linear-time algorithms (in $n$ but not in $n^2$) are often enough for large-scale applications, provided that powerful hardware is available. However, we believe that constant-time algorithms can make it possible for more researchers in the community to study large-scale applications, which would contribute to potential research outcomes. Currently, processing large scales of data is usually used exclusively by leading internet companies like Google and Microsoft. In contrast, our experiments are conducted on a personal computer with an Intel Core i9 CPU of 32GB memory, and we can already scale to $10^5$ vertices ($10^9$ scale of edges). By significantly lowering the computational barrier, our algorithm allows large-scale analysis to be performed on common hardware, opening the door to new applications previously hindered by resource constraints.

---

### Official Review · Reviewer_kSQi · 2025-11-01

**Soundness:** 3
**Presentation:** 3
**Contribution:** 2
**Rating:** 6
**Confidence:** 3

**Summary:**

This paper studies property testing for correlation clustering and structural balance in dense labeled graphs. It improves the known query complexity, and achieves a tight bound for the two-cluster (balance) case. Novel results, for the $k$ clusters case are also given. The improvement comes primarily from a cleaner probabilistic analysis using Janson’s inequality, rather than from a new algorithmic idea.

**Strengths:**

- **Analytical improvement over prior bounds.**
The work reduces the query complexity for testing correlation clustering from $\tilde{O}(1/\varepsilon^7)$ to $O(1/\varepsilon^2)$, and achieves a tight $\Theta(1/\varepsilon)$ bound for the special case $k=2$. Although the improvement stems mainly from a sharper analysis, the result is still valuable as it narrow an asymptotic gap. Furthermore, the application of Janson’s inequality to handle dependent random variables is mathematically clean and simplifies the analysis. While not conceptually deep, it removes several unnecessary factors and clarifies the dependence on $\varepsilon$. Finally, the case for $k$-clusters seems novel.

- **Theoretically relevant within property testing.**
Within the dense-graph property testing framework, correlation clustering is a natural and well-motivated property. The paper contributes a clearer understanding of its sample complexity, which is of interest to the theory community.

**Weaknesses:**

- **Primarily analytical, not algorithmic, contribution.**
The main improvement derives from a tighter analysis rather than a novel testing algorithm. The core testing procedure remains essentially unchanged from prior work, and the use of Janson’s inequality, while effective, is a straightforward application of a standard concentration tool.

- **Limited depth of the probabilistic insight.**
The argument simplifies the dependency structure among sampled edges but does not introduce new combinatorial or probabilistic ideas. The simplicity of the improvement raises the question of why such a refinement was not already known.

- **Coarse granularity of the testing objective.**
As in all property testing formulations, the test only distinguishes between perfectly clusterable graphs ($\mathrm{OPT}=0$) and those that are $\varepsilon$-far from such structure ($\mathrm{OPT} \ge \varepsilon \binom{n}{2}$). This dichotomy is extremely coarse and offers little interpretive value for practical clustering, where intermediate cases are the norm.

- **Relevance limited to dense theoretical settings.**
The dense graph model and the assumption of oracle access to edge labels make the results largely theoretical. The test loses meaning when $\varepsilon = O(1/n)$, since the query complexity becomes $\Theta(n^2)$, equivalent to reading the entire input. The authors do not discuss this limitation. Moreover, while the $k$-clusters case is novel the $k^4 \log^4 k$ runtime seems problematic.

- **Moderate originality.**
The methodological novelty is incremental, and the improvement, though useful, does not introduce a fundamentally new perspective or analysis technique.

### Overall evaluation

This is a technically solid and clearly written paper that refines the analysis of property testing for correlation clustering. The results are correct, and the improvements close an asymptotic gap in query complexity, but the novelty is mostly analytical rather than conceptual or algorithmic. The paper will interest the property testing and theoretical graph learning communities, though its impact on broader learning theory or clustering practice is limited.

**Questions:**

See Weaknesses section.

---

> ### Author Response · Authors · 2025-11-24
>
> We thank the reviewer for the careful review and the positive feedback. Our responses to your weakness comments and questions are as follows.
>
> ### **Response to “Primarily analytical, not algorithmic, contribution” and “Limited depth of the probabilistic insight”**
>
> We combine the answers to these two questions since they have significant overlaps.
>
> We agree with the reviewer that the algorithm is relatively straightforward. However, we would like to refer to a common understanding of property testing problems, raised by prior works such as Goldreich and Trevisan [FOCS’01]: **Most graph property testing problems admit ‘canonical testers’**, i.e., an algorithm that samples a certain number of vertices from $V$, tests the property on the sampled subset, and returns the answer accordingly. Therefore the key technical challenge falls into a tighter analysis. Our algorithms for testing correlation clustering indeed employs this canonical tester with fewer samples, and our analysis tackles the technical barrier by new insights and the use of Janson’s inequality.
>
> Furthermore, we believe simplicity is a strength rather than a weakness. For modern large-scale algorithms in ML, simplicity ensures that the algorithm can be easily implemented by practitioners, and the provable guarantees ensure the quality of the solution. Our algorithms satisfy both properties.
>
> Finally, we remark that our analysis is non-trivial and requires strong technical insights. The reason for the improvement not being known before is mainly because previous work has not built the insight we explored in the paper. Therefore, we believe this further validates our technical contribution and strength.
>
> **Revision action**: In the revised version of the paper, we have added a discussion about the simplicity of our algorithm, the canonical tester, and the main reason our results are not known before (line 138).
>
> Reference: Goldreich and Trevisan [FOCS’01] “Three Theorems Regarding Testing Graph Properties”
>
> ### **Response to “Coarse granularity of the testing objective”**
>
> Thank you for pointing out the concern. We would like to clarify that our algorithms are actually tolerant testers, i.e., $\text{OPT}$ being $\alpha$-close to $0$ vs. $\beta$-far from $0$.
>
> - For structural balance, in Theorem 4 (details in Appendix F), we proved that our algorithm is able to distinguish graphs that are at most $\delta$-close to being balanced vs. at least $\varepsilon$-far from being balanced for $\delta=C\cdot \varepsilon$ for some constant $C<1$.
>
> - For general correlation clustering, our algorithm can distinguish graphs that are $\varepsilon$-far from being clusterable vs. $O(\varepsilon^2)$-close to being clusterable, although we did not explicitly state it due to the context organization. Using the same proof idea as we did for the proof of Theorem 2 (between lines 355 and 365 of the original version), we can show that if the graph is at most $\varepsilon^2$-close from being balanced, then no flipped edge will be sampled, which means the algorithm will return ‘YES’ in such a case.
>
>
> **Revision action**: In the updated version of the paper, we have updated our statement and the proof for the testing algorithm for clusterability.
>
>
> ### **Response to “Relevance limited to dense theoretical settings”**
>
> Thank you for the comments. We remark that the original problems of correlation clustering and structural balance are indeed defined on dense graphs. In practice, for an input graph $G=(V,E)$ not necessarily complete, we can run the algorithm by treating the edges as $(+)$ edges and the non-edges as $(-)$ ones. Although this strategy does not fit all application scenarios, for applications like social networks, it makes perfect sense ($u,v$ are friends if they are connected, and not friends otherwise). In our experiment, we used this strategy to obtain good performances, which validates the approach.
>
> For the concern of small $\varepsilon$ values: usually, we are not interested in $\varepsilon=O(1/n)$ for property testing questions. In Appendix G of our paper, we proved that the problem requires at least $\Omega(1/\varepsilon)$ samples. Therefore, if the value of $\varepsilon$ is so small, any algorithm would need more than $n$ queries.

---

> > ### Author Response · Authors · 2025-11-24
> >
> > ### **Response to “Moderate originality”**
> >
> > We respectfully disagree with this statement. Please check the rationale below.
> >
> > - First, as we discussed earlier, prior work (Goldreich and Trevisan [FOCS’01]) has pointed out most graph property testing problems admit a “canonical tester”. Therefore the key technical challenge usually falls into analysis.
> > - Our paper presents a novel analysis of this canonical tester, which has a fundamentally different structure from existing ones: we analyze the probability of sampling positive paths and apply Janson’s inequality.
> > - Our analysis established a new approach, which may open new opportunities. In the analysis of related problems, such as subgraph freeness testing, the analysis relies on the graph removal lemma, which is too coarse to get a strong bound with a small exponent. In the prior work (Adriaens and Apers, 2023), the reduction to the MAX-CSP does not yield a tight bound either: existing bounds on MAX-CSP have a high exponent in $1/\varepsilon$.
> >
> > Therefore, our analysis is original and becomes the key to obtaining these results.

---

> > > ### Comment · Reviewer_kSQi · 2025-11-27
> > >
> > > I thank the authors for the detailed rebuttal and for clarifying the originality of their analysis.
> > >
> > > My main remaining concern is the relevance of this contribution to the broader ML community. While the proposed analysis indeed represents a substantial improvement over prior work on this specific tester, I am not fully convinced that results of this nature translate into clear benefits for practical clustering problems. This concern is further reinforced by the high computational complexity of the method, particularly the runtime for the setting of arbitrary k.
> > >
> > > Overall, I consider this a technically sound contribution (and my score is indeed 6). However, in my view, it does not rise to the level of the strongest submissions in terms of relevance to mainstream machine learning settings.

---

### Author Response · Authors · 2025-11-24
**Rebuttal and an Updated Version of Our Paper**

We are grateful for the insightful reviews provided by the reviewers.

We have addressed each question in the posted rebuttal, and we have revised the manuscript according to the reviewers' valuable feedback.

We hope the reviewers find our responses and revisions satisfactory for the questions and concerns. Also, please let us know if there are further questions.

Best regards,
Authors

---

### Author Response · Authors · 2025-12-02
**Author Final Remarks**

Dear Reviewers and AC,

We sincerely thank the reviewers for the insightful reviews and helpful suggestions. We want to add some final remarks on the merits of the paper and how we have addressed all the concerns.


## Reviewers have pointed out multiple merits of our paper

- Our problem is well-motivated and well-formulated (Reviewers kSQi, Yvv8, and ENS9)

- Our theoretical results are solid and non-trivial (All reviewers)

- Our improvement over the previous best algorithm is significant (Reviewers kSQi and Yvv8)

- Our usage of Janson’s inequality on property testing tasks is interesting (Reviewers CjKd and Yvv8)

- Our experimental results validate the theoretical claims (Reviewers ENS9 and Yvv8)

- The paper is well-written (Reviewers kSQi, ENS9 and Yvv8)


## We have addressed all the concerns

We have responded to the reviewers about their concerns and revised the manuscript based on their suggestions. In particular, we have addressed the following concerns:

- Novelty of our algorithm: We have included a discussion about ‘canonical testers’ in property testing problems. As such, the main novelty for papers in property testing lies exactly in tighter analysis, which is what we did in the paper.

- Tolerate testing: we have clarified that our algorithms can indeed perform tolerant testing, and we have added the updated proofs to the paper.

- Complete graph assumption and practical concerns: we have clarified that in many practical cases, we can generate the $(-)$ edges simply from the non-edges in the graphs. Furthermore, we emphasized that this strategy performs well in our experiments.

- Clarity and writing suggestions: We thank the reviewers for pointing out the writing concerns, and we have applied all the comments from the reviewers in the updated version.

---

### Meta-Review · Area_Chair_Qccj · 2026-01-12

**Summary:**

The majority of the reviewers vote for acceptance. Reviewer ENS9 is against it, however, their concern is addressed.

**Reviewer Concerns:**

Main concern

> W1. While O(1/epsilon^2) is a significant improvement over \tilde{O}(1/epsilon^7), I found that the two results are not directly comparable because the earlier work of AA2023 tests of cost smaller than epsilon/10, while the current work tests for cost=0. So, I think that the current paper solves a much easier problem. Furthermore, this limitation is significant in practice, as the method essentially does not tolerate any noise, which is highly unrealistic for real-world applications. I think that the author(s) should have been more clear about this important distinction, during the discussion and when presenting Table 1. This weakness is my main motivation for not suggesting the paper for acceptance.

No other outstanding concerns.

**Reviewer Scores:**

Sorry, I am not able to provide this. That's speculation.

---

### Decision · Program_Chairs · 2026-01-26

Accept (Poster)